# RSA-CP: Efficient Conformal Prediction in Small-Sample Regimes via Random Score Alignment

**Pankaj Bhagwat** [1]  **Zhixian Yang** [1]  **Yihao Wang** [1]  **Bei Jiang** [1]  **Linglong Kong** [1]

## Abstract

Conformal Prediction (CP) provides rigorous finite-sample coverage guarantees, yet its statistical efficiency depends critically on the size of the calibration set. In data-scarce regimes, CP often suffers from volatile quantile estimation, leading to overly conservative and wide prediction intervals. To address this, we propose Random Score Alignment-Conformal Prediction (RSA-CP), a simple framework designed to improve sample efficiency in small-sample CP. Instead of requiring the computationally intensive generation of full synthetic datasets, RSA-CP enhances calibration by directly aligning real scores with a high-resolution reference score distribution. By employing an optimal transport mapping, our framework refines the stepwise quantile increments through a globally optimal use of reference information. We provide theoretical guarantees establishing that RSA-CP maintains robust coverage without any distributional assumptions on the reference scores. Empirical evaluations demonstrate that RSA-CP consistently produces more efficient prediction sets while maintaining calibrated coverage behavior. Overall, RSA-CP offers a computationally efficient and theoretically grounded solution for robust uncertainty quantification under limited data.

## 1. Introduction

Reliable uncertainty quantification is a prerequisite for the safe deployment of machine learning systems in high-stakes domains, including healthcare (Esteva et al., 2017), autonomous systems (Bojarski et al., 2016), and legal decision-making. Conformal prediction (CP) (Vovk et al., 2005; Saunders et al., 1999; Lei et al., 2018; Angelopoulos & Bates, 2021) provides a principled framework for this task by offering finite-sample, distribution-free coverage guarantees under minimal assumptions. Given a test input with an unknown label, CP constructs a prediction set guaranteed to contain the true label with a user-specified probability (e.g., 95%), using nonconformity scores computed on a labeled calibration set. These guarantees rely only on exchangeability between calibration and test samples and do not require assumptions on the underlying data-generating distribution.

Despite its strong theoretical guarantees, the practical performance of CP deteriorates sharply when calibration data are scarce. In small-sample regimes, prediction sets often become overly conservative, limiting their usefulness for applications that require personalization or subgroup-specific guarantees (Vadlamani et al., 2025; Lu et al., 2022; Gibbs et al., 2025). This challenge is particularly pronounced in medical diagnostics (Banerji et al., 2023; Liu & Meng, 2016) and class-conditional image classification (Vovk et al., 2003; Ding et al., 2023), where only a limited number of labeled examples may be available per group. At a technical level, this limitation arises from a fundamental *resolution bottleneck*: the empirical distribution of conformity ranks is discrete, preventing fine-grained control of miscoverage when the target level $\alpha$ is small relative to $1/(m+1)$. To preserve validity under these conditions, standard CP procedures may be forced to adopt vacuous thresholds, resulting in prediction sets that effectively cover the entire label space. While such sets are mathematically valid, they provide little practical value for decision-making.

Recent work has proposed a variety of mechanisms to mitigate the small-sample limitations of CP, including the use of synthetic data (Bashari et al., 2026), clustering-based calibration (Gao et al., 2025a), posterior conformal prediction (Zhang & Candès, 2024), and related approaches (see Appendix A). A common theme underlying these methods is the augmentation of the calibration scores with additional data-derived scores, compensating for the scarcity of real calibration samples. However, a fundamental challenge arises: the distribution of the augmented data is generally not guaranteed to match that of the true calibration data. As a result, finite-sample coverage guarantees, the defin-

---

[1]Department of Mathematical and Statistical Sciences, University of Alberta, Edmonton, Canada. Correspondence to: Pankaj Bhagwat <pbhagwat@ualberta.ca>.

*Proceedings of the 43rd International Conference on Machine Learning*, Seoul, South Korea. PMLR 306, 2026. Copyright 2026 by the author(s).

ing strength of the CP framework, are typically lost unless strong assumptions are imposed, such as exact distributional equivalence or explicit bounds on distributional mismatch. An exception is Bashari et al.'s (2025) synthetic-prediction powered inference (SPI) framework, which successfully bypasses this issue by carefully controlling the augmentation mechanism. Nevertheless, SPI relies on access to reliable synthetic data generation procedures, which may be unavailable in many high-stakes applications and can be computationally expensive to train and deploy at scale.

In this work, we take a different perspective. Rather than focusing on how to generate additional data, we ask what *properties* augmented scores must satisfy in order to preserve finite-sample validity. Guided by this analysis, we introduce a universally applicable and computationally lightweight framework that avoids reliance on data generation altogether. Our key observation is that nonconformity scores can be *aligned* with scores drawn from a reference distribution, enabling principled calibration even when real calibration data are scarce. This insight leads to a new conformal mechanism, which we term *Random Score Alignment Conformal Prediction* (RSA-CP). We establish explicit finite-sample coverage bounds for prediction sets constructed under this framework and show how score alignment can be exploited to improve the volume efficiency of the resulting sets.

**Our Contributions:**

- **Reference-score-augmented conformal prediction:** We propose a new conformal prediction framework, RSA-CP, that leverages *reference scores*, which may be random draws from a known distribution, to improve the resolution of split conformal calibration and construct informative prediction sets in small-sample regimes, where standard conformal methods often produce overly conservative or even vacuous sets.

- **Rank-resolution view of score augmentation:** We characterize the augmented rank of the test score via a conditional Beta-Binomial law, leading to high-probability conditional rank windows that quantify how additional scores refine the coarse rank grid induced by small $m$.

- **Distribution-free finite-sample coverage bounds:** We derive distribution-free finite-sample coverage bounds for prediction sets constructed using RSA-CP. These bounds are easily computable and can be evaluated independently of the observed calibration data.

- **Rank-preserving alignment and distribution-robust refinement.** We incorporate monotone score alignment maps (including an OT barycentric alignment) that inherits the distribution-free bounds, and we further derive distribution-robust coverage guarantees

that tighten around the target coverage level when the aligned real score distribution is close to the reference distribution.

Overall, our results show that in small-sample regimes, existing conformal methods either fail to provide informative prediction sets or rely on strong modeling assumptions. Recent approaches based on synthetic data generation provide distribution-free guarantees but require access to complex generators that may be computationally expensive, application-specific, or unavailable in high-stakes settings. In contrast, RSA-CP achieves improved calibration resolution and informative prediction sets using only reference scores, while retaining principled finite-sample coverage bounds.

**Conflict of Interest Disclosure.** The authors declare no financial conflicts of interest related to this work. In particular, this paper does not evaluate a model, product, or system developed by an organization that employs any of the authors.

## 2. Background

We consider the problem of distribution-free predictive uncertainty quantification. Let $(X, Y) \in \mathcal{X} \times \mathcal{Y}$ be a random covariate-response pair drawn from an unknown distribution $P$. For a new covariate realization $X_{m+1}$, the objective is to output a set-valued prediction rule $\widehat{C}(X_{m+1}) \subseteq \mathcal{Y}$ that satisfies the marginal coverage constraint

$$\mathbb{P}\left\{Y_{m+1} \in \widehat{C}(X_{m+1})\right\} \geq 1 - \alpha, \qquad (1)$$

for a user-specified miscoverage level $\alpha \in (0, 1)$, where $(X_{m+1}, Y_{m+1}) \sim P$ is an independent test draw. To construct such prediction sets, we are given a labeled calibration sample $\mathcal{D}_{\text{real}} = \{(X_i, Y_i)\}_{i=1}^m$, whose elements are exchangeable with the test point. Exchangeability is the only structural assumption required and forms the theoretical foundation of CP. Under this assumption, conformal methods deliver exact finite-sample coverage guarantees in (1) without requiring parametric or distributional knowledge of $P$.

CP operates by transforming labeled examples into real-valued *nonconformity scores*. We assume access to a fixed scoring function $s : \mathcal{X} \times \mathcal{Y} \rightarrow \mathbb{R}$, trained independently of the calibration data, which assigns larger values to less plausible input–label pairs. Typical choices include absolute prediction errors, inverse class probabilities, or negative log-likelihoods. Applying $s$ to the calibration sample yields scores $S_i = s(X_i, Y_i)$, $i = 1, \ldots, m$, while the corresponding score $S_{m+1} = s(X_{m+1}, Y_{m+1})$ for the test example remains unobserved at prediction time.

## 2.1. Split Conformal Prediction (SCP)

Split conformal prediction (SCP) is the simplest and most widely adopted conformal method, constructing prediction sets via score thresholding (Vovk et al., 2005; Papadopoulos et al., 2002). Let $S_{(1)} \leq S_{(2)} \leq \cdots \leq S_{(m)}$ denote the ordered calibration scores. For a target miscoverage level $\alpha$, SCP defines the threshold $\hat{q}_{\mathrm{SCP}} := S_{(k)}, k = \lceil (m+1)(1-\alpha) \rceil$, and outputs the prediction set

$$\widehat{C}_{\mathrm{SCP}}(X_{m+1}) = \{y \in \mathcal{Y} : s(X_{m+1}, y) \leq \hat{q}_{\mathrm{SCP}}\}.$$

Under the exchangeability assumption, the rank of $S_{m+1}$ among the multiset $\{S_1, \ldots, S_m, S_{m+1}\}$ is uniformly distributed over $\{1, \ldots, m+1\}$. This property immediately implies that $\widehat{C}_{\mathrm{SCP}}$ satisfies the marginal coverage guarantee in (1). Notably, this validity (1) holds for arbitrary score functions and for any underlying data-generating distribution $P$. We also have, if scores are distinct (Vovk et al., 2005; Papadopoulos et al., 2002; Angelopoulos & Bates, 2021),

$$\left| \mathbb{P}\{Y_{m+1} \in \widehat{C}_{\mathrm{SCP}}(X_{m+1})\} - (1-\alpha) \right| \leq \frac{1}{m+1}. \quad (2)$$

## 2.2. A Fundamental Limitation: Instability in Low-Calibration Regimes

CP guarantees exact finite-sample validity, but its efficiency is highly sensitive to the calibration sample size. When the number of calibration points $m$ is small, the empirical distribution of nonconformity scores becomes coarse and unstable (see eq. (2)). In particular, a few atypical calibration scores can dominate the empirical $(1-\alpha)$ quantile, substantially inflating the calibration threshold.

This effect leads to overly conservative prediction sets that are much wider than necessary to achieve coverage. In extreme cases, the resulting sets may cover a large portion of the output space, offering limited practical utility despite formally satisfying the coverage guarantee. To illustrate the small-sample instability of SCP, we simulated calibration scores from a Gamma distribution and examined the distribution of conformal interval lengths across 1000 repetitions (see Figure 1). For target coverage $1-\alpha = 0.95$ and calibration size $m = 20$, SCP uses the maximum calibration score as the conformal threshold, resulting in highly variable interval lengths. In contrast, when $m = 2000$, the conformal threshold corresponds to a stable interior quantile and the interval-length distribution becomes sharply concentrated. This phenomenon highlights the coarse rank resolution problem in small-sample conformal prediction.

Addressing this instability, without sacrificing finite-sample coverage guarantees, is therefore essential for deploying CP in low-calibration regimes.

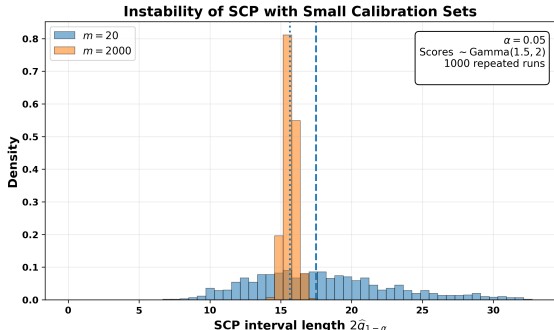

*Figure 1.* Illustration of the instability of SCP under small calibration sizes. Calibration scores were generated from a Gamma distribution, and the resulting SCP interval lengths were computed across 1000 repetitions for target coverage $1 - \alpha = 0.95$. When the calibration size is small ($m = 20$), SCP uses an extreme order statistic as the conformal threshold, producing highly variable and often overly conservative interval lengths. In contrast, for large calibration sizes ($m = 2000$), the threshold corresponds to a stable interior quantile and the interval-length distribution becomes sharply concentrated. This phenomenon illustrates the coarse rank resolution problem that motivates our work.

## 3. Our Framework

Our objective is to improve the sample efficiency of SCP in small-data settings by leveraging augmented nonconformity scores, while retaining finite-sample coverage bounds of the form (1). Since SCP depends entirely on the empirical distribution of calibration scores, its inefficiency in small samples is fundamentally a consequence of limited quantile resolution.

To isolate this effect, consider an idealized setting in which, in addition to the $m$ observed calibration scores $\{S_1, \ldots, S_m\}$ drawn i.i.d. from the true score distribution induced by $P$ (assumed continuous on $\mathbb{R}$), we have access to $N$ additional i.i.d. scores $\{\tilde{S}_1, \ldots, \tilde{S}_N\}$ from the same distribution. Applying SCP to the augmented collection $\{S_1, \ldots, S_m\} \cup \{\tilde{S}_1, \ldots, \tilde{S}_N\} \cup \{S_{m+1}\}$ yields a prediction set $\widehat{C}_{\mathrm{SCP}, m+1, N}$ satisfying

$$\left| \mathbb{P}\left\{ Y_{m+1} \in \widehat{C}_{\mathrm{SCP}}(X_{m+1}) \right\} - (1-\alpha) \right| \leq \frac{1}{m+N+1}.$$

This highlights a central insight: augmenting calibration scores increases the effective resolution of the empirical rank distribution from $1/(m+1)$ to $1/(m+N+1)$. Equivalently, the attainable precision of the calibration quantile improves at rate $\mathcal{O}(1/N)$, leading to a more stable threshold and, consequently, narrower and more reliable prediction sets.

Of course, in practice, additional scores drawn from the true score distribution are unavailable. We instead propose to augment the $m$ real calibration scores with *reference scores* sampled from a user-chosen distribution. These reference scores need not match the true score distribution of the real

calibration scores. This raises the central question of the paper:

> *When can we use reference scores to increase calibration resolution and still retain finite-sample coverage?*

In the remainder of the section, we outline the strategy to address this question.

### 3.1. A Rank-Based View of Calibration Resolution

We begin by analyzing conformal calibration under score augmentation in an idealized setting, where additional scores are drawn from the same distribution as the real calibration scores. Let $r_{m+1,m}$ denote the rank of $S_{m+1}$ among $\{S_1, \ldots, S_m, S_{m+1}\}$. By exchangeability,

$$\mathbb{P}(r_{m+1,m} = k) = \frac{1}{m+1}, \qquad k = 1, \ldots, m+1. \quad (3)$$

Now augment the calibration set with $N$ additional scores $\tilde{S}_1, \ldots, \tilde{S}_N$, drawn i.i.d. from the same continuous distribution and independent of the original scores. Let $r_{m+1,m+N+1}$ denote the rank of $S_{m+1}$ in the augmented set of size $m + N + 1$.

Define $B := \sum_{j=1}^{N} \mathbf{1}\{\tilde{S}_j \leq S_{m+1}\}$, the number of augmented scores not exceeding the test score. Since the distribution is continuous, ties occur with probability zero, and $r_{m+1,m+N+1} = k + B$.

Conditionally on the event $\{r_{m+1,m} = k\}$, the random variable $B$ follows a Beta-Binomial distribution (see Appendix C.1 for technical details), for $b = 0, 1, \ldots, N$,

$$\mathbb{P}(B = b \mid r_{m+1,m} = k) =$$
$$\binom{N}{b} \frac{\mathcal{B}(b + k, \ N - b + m + 2 - k)}{\mathcal{B}(k, \ m + 2 - k)},$$

where $\mathcal{B}(\cdot, \cdot)$ denotes the Beta function (Eq. (12)). Equivalently, for $t \in \{k, k+1, \ldots, k+N\}$,

$$\mathbb{P}(r_{m+1,m+N+1} = t \mid r_{m+1,m} = k) =$$
$$\binom{N}{t-k} \frac{\mathcal{B}(t, \ m + N + 2 - t)}{\mathcal{B}(k, \ m + 2 - k)}. \quad (4)$$

The detailed proof is provided in the Appendix C.1.1.

### 3.2. High-Probability Conditional Rank Windows

The conditional rank distribution derived in (4) provides a principled way to localize the augmented rank of the test score, given its rank among the real calibration scores. We now leverage this structure to construct *high-probability conditional rank windows*.

Fix a rank $k \in \{1, \ldots, m+1\}$ and consider the conditional distribution of the augmented rank $r_{m+1,m+N+1}$ given $\{r_{m+1,m} = k\}$. By (4), this distribution is supported on $\{k, \ldots, k+N\}$ and is fully characterized by the Beta-Binomial law. For a user-specified confidence level $1 - \beta \in (0, 1)$, we define integers

$$b_-(k, \beta) := Q_{B|k}\left(\tfrac{\beta}{2}\right), \qquad b_+(k, \beta) := Q_{B|k}\left(1 - \tfrac{\beta}{2}\right),$$

where $Q_{B|k}$ denotes the conditional quantile function of $B$ given $r_{m+1,m} = k$. Equivalently, with probability at least $1 - \beta$,

$$r_{m+1,m+N+1} \in \left[k + b_-(k, \beta), \ k + b_+(k, \beta)\right].$$

These conditional rank windows capture, with high probability, the refined location of the test score rank within the augmented set, conditional on its coarse rank among the real calibration scores.

### 3.3. Prediction Sets via Conditional Rank Compatibility

We now translate the high-probability conditional rank windows into a conformal prediction set. The key idea is to include a candidate label $y$ whenever the nominal conformal cutoff is *compatible* with the conditional rank window implied by the real calibration scores.

Fix a candidate label $y$ and let $k := r_{m+1,m}(y)$ denote the rank of its nonconformity score among the $m$ real calibration scores. Let

$$j^* := \lceil (1 - \alpha)(N + m + 1) \rceil$$

denote the nominal conformal cutoff corresponding to target miscoverage level $\alpha$ on the augmented rank scale. We include $y$ in the prediction set whenever this cutoff is compatible with the conditional rank window, that is,

$$y \in \widehat{C}(X_{m+1}) \iff k + b_+(k, \beta) \leq j^* \quad \text{or}$$
$$k + b_-(k, \beta) \leq j^* \leq k + b_+(k, \beta). \quad (5)$$

Concretely, inclusion occurs when the entire conditional rank window lies below the cutoff or when the cutoff falls inside the conditional rank window.

We now establish finite-sample coverage guarantees for the proposed prediction sets. The key observation is that the rank-based inclusion rule in (5) admits deterministic lower and upper bounds expressed through indicator functions of the underlying rank.

**Theorem 3.1** (Finite-sample coverage bounds)**.** *For any*

$\alpha, \beta \in (0,1)$ *and $j^\star$ be as above. Let*

$$L(\beta, m) = \frac{1}{m+1} \sum_{k=1}^{m+1} \mathbf{1}\{ k + b_+(k, \beta) \le j^\star \},$$

$$U(\beta, m) = \frac{1}{m+1} \sum_{k=1}^{m+1} \mathbf{1}\{ k + b_-(k, \beta) \le j^\star \}.$$

*Then,*

$$L(\beta, m) \;\le\; \mathbb{P}\Big(Y_{m+1} \in \widehat{C}(X_{m+1})\Big) \;\le\; U(\beta, m). \quad (6)$$

The lower bound $L(\beta, m)$ counts the fraction of real-score ranks for which the entire conditional rank window lies below the conformal cutoff. The upper bound $U(\beta, m)$ counts the fraction of ranks for which the conditional window intersects or lies below the cutoff. The gap between these bounds reflects the residual uncertainty induced by conditional rank localization.

Importantly, the bounds in Theorem 3.1 depend only on the combinatorial rank quantities $b_-(k, \beta)$ and $b_+(k, \beta)$ and make no assumptions about the true score distribution induced by $P$. Moreover, the inclusion rule in (5) is formulated entirely in terms of ranks and Beta-Binomial thresholds, without reference to the numerical values of the scores themselves. This rank-based formulation is central: it allows the decision rule to be mapped back to score space via an arbitrary quantile map.

### 3.4. Mapping the inclusion rule back to score space

A key feature of the inclusion rule in (5) is that it is defined entirely in terms of *ranks*, and is therefore independent of the numerical values of the scores. Consequently, the rule can be transported to an arbitrary score space through a user-chosen reference distribution, without affecting the finite-sample guarantees in Theorem 3.1.

Let $\mathcal{A} = \{S_1, \ldots, S_m\} \cup \{\widetilde{S}_1, \ldots, \widetilde{S}_N\}$ denote the augmented score collection, and let $A_{(1)} \le A_{(2)} \le \cdots \le A_{(m+N)}$ be its order statistics, with the conventions $A_{(0)} = -\infty$ and $A_{(m+N+1)} = +\infty$. Then the inclusion rule admits the explicit score-threshold form

$$y \in \widehat{C}_{RSA}(X_{m+1}) \iff$$

$$s(X_{m+1}, y) \;\le\; \min\Big\{ A_{\big(k+b_+(k,\beta)\big)}, \; A_{(j^\star)} \Big\} \quad \text{or}$$

$$s(X_{m+1}, y) \;\le\; A_{\big(k+b_-(k,\beta)\big)}. \quad (7)$$

We call $\widehat{C}_{RSA}(X_{m+1})$ Random Score Alignment- Conformal prediction (RSA-CP) sets.

**Proposition 3.2** (Score-space implementation preserves the bounds). *Let $\tilde{Q}$ be the empirical CDF of the reference scores*

$\{\tilde{S}_1, \ldots, \tilde{S}_N\}$ *with generalized inverse $\tilde{Q}^{-1}$. Then prediction sets in* (5) *and* (7) *are equal almost surely. In particular, $\widehat{C}_{RSA}$ satisfies the same finite-sample coverage bounds as in Theorem 3.1.*

Proposition 3.2 justifies implementing the inclusion rule directly in score space using the reference score order statistics, leading to the practical prediction-set construction summarized in Algorithm 1.

---

**Algorithm 1** RSA-CP Prediction Set Construction

---

1: **Input:** Calibration data $\{(X_i, Y_i)\}_{i=1}^m$, score function $s(\cdot, \cdot)$, reference scores $\{\widetilde{S}_j\}_{j=1}^N$, miscoverage level $\alpha \in (0,1)$, window level $\beta \in (0,1)$.

2: **Output:** Prediction set $\widehat{C}_{\mathrm{RSA}}(X_{m+1}) \subseteq \mathcal{Y}$ for test input $X_{m+1}$.

3: Compute real calibration scores $S_i \leftarrow s(X_i, Y_i)$ for $i = 1, \ldots, m$.

4: Form the augmented score collection $\mathcal{A} = \{S_1, \ldots, S_m\} \cup \{\widetilde{S}_1, \ldots, \widetilde{S}_N\}$.

5: Sort the augmented scores: $A_{(1)} \le A_{(2)} \le \cdots \le A_{(m+N)}$, with conventions $A_{(0)} = -\infty$ and $A_{(m+N+1)} = +\infty$.

6: Set cutoff index $j^\star \leftarrow \lceil (1-\alpha)(m+N+1) \rceil$.

7: Initialize $\widehat{C}_{\mathrm{RSA}}(X_{m+1}) \leftarrow \emptyset$.

8: **for** each candidate label $y \in \mathcal{Y}$ **do**

9:     Compute candidate score $s_y \leftarrow s(X_{m+1}, y)$.

10:     Compute the real-score rank $k \leftarrow r_{m+1,m}(s_y) = 1 + \sum_{i=1}^m \mathbf{1}\{S_i \le s_y\}$.

11:     Compute Beta-Binomial window indices $b_- \leftarrow b_-(k, \beta)$, $b_+ \leftarrow b_+(k, \beta)$, where $B \mid k \sim \mathrm{BetaBin}(N, k, m+2-k)$.

12:     Set the RSA-CP score threshold $q_{\mathrm{RSA}}(k) \leftarrow \max\{\min(A_{(k+b_+)}, A_{(j^\star)}), A_{(k+b_-)}\}$.

13:     **if** $s_y \le q_{\mathrm{RSA}}(k)$ **then**

14:         $\widehat{C}_{\mathrm{RSA}}(X_{m+1}) \leftarrow \widehat{C}_{\mathrm{RSA}}(X_{m+1}) \cup \{y\}$.

15:     **end if**

16: **end for**

---

### 3.5. Choosing $\beta$ for Target Coverage

The finite-sample bounds in Theorem 3.1 depend on the window level $\beta$, which controls the width of the conditional rank window. Smaller values of $\beta$ produce wider windows and more conservative prediction sets, while larger values yield narrower windows and improved efficiency.

A natural objective is to select $\beta$ so that the prediction set defined by the inclusion rule achieves the target marginal coverage level $1-\alpha$. Since the lower bound in Theorem 3.1 is distribution-free, a sufficient condition for validity is

$$L(\beta, m) \;\ge\; 1 - \alpha. \quad (8)$$

When this condition holds, the resulting prediction set is guaranteed to attain at least the desired coverage. Algorithm 2 selects the largest $\beta$ satisfying (8), thereby maximizing efficiency subject to certified validity (finite-sample coverage bounds in Theorem 3.1). This search is well posed because $L(\beta, m)$ is nondecreasing in $\beta$ (Lemma C.2), while the corresponding upper bound $U(\beta, m)$ is nonincreasing (Lemma C.1).

Due to rank discreteness and the conservativeness of distribution-free lower bounds, it is possible that (8) fails for all $\beta \in (0, 1)$, even when the true coverage is close to the target level. This reflects a fundamental limitation of certification under coarse rank resolution, rather than a deficiency of the inclusion rule itself.

In the next section, we show how to move beyond lower-bound certification to construct narrower prediction sets while achieving near to target coverage levels.

### 3.6. Score Alignment Step

Although Theorem 3.1 provides finite-sample, distribution-free coverage bounds for the prediction set defined by the inclusion rule, the certified lower bound $L(\beta, m)$ may fall well below the target level $1 - \alpha$ for all choices of $\beta \in (0, 1)$. This behavior does not reflect a failure of the inclusion rule itself, but rather a mismatch between the real calibration scores and the reference score system used to define the cutoff.

From the inclusion rule (7), the procedure falls back to the lower bound $L(\beta, m)$, if $\tilde{S}_{\left(k+b_+(k,\beta)\right)} \geq \tilde{S}_{\left(j^*\right)}$ a.s. Conversely, poor alignment between the real and reference scores (e.g., differing supports, scales, or distributional shapes) can make this fallback very often.

To address this issue, we introduce a *score alignment* step whose goal is to improve coverage within the certified bounds by enhancing compatibility between the real and reference score systems. Importantly, any such alignment must preserve the relative ordering of the real scores, since all finite-sample guarantees rely exclusively on rank information.

#### 3.6.1. RANK-PRESERVING SCORE ALIGNMENT.

We therefore consider transformations $T : \mathbb{R} \longrightarrow \mathbb{R}$, that are nondecreasing, so that $s \leq s'$ implies $T(s) \leq T(s')$. Such transformations preserve the ranks of the real scores while allowing their numerical scale and support to be adjusted.

Applying the inclusion rule to the aligned scores $\{T(S_i)\}$ and $T(s(X_{m+1}, y))$ leaves the rank-based structure of the method unchanged and hence preserves all finite-sample guarantees, while potentially improving the interaction be-

tween real and reference scores. In the next section, we construct principled alignment maps and show that appropriate alignment can substantially tighten the coverage bounds and yield more informative prediction sets. Consider $\mathcal{A}_T = \{T(S_1), \dots, T(S_m)\} \cup \{\widetilde{S}_1, \dots, \widetilde{S}_N\}$, with order statistics $A^T_{(1)} \leq \cdots \leq A^T_{(m+N)}$.

**Proposition 3.3** (Aligned Score-space implementation preserves the bounds). *Let $\widehat{C}_{RSA,T}(X_{m+1})$ defined via the inclusion rule:*

$$
y \in \widehat{C}_{RSA,T}(X_{m+1}) \iff
$$
$$
T(s(X_{m+1}, y)) \leq \min\left\{ A^T_{\left(k+b_+(k,\beta)\right)}, \ A^T_{\left(j^\star\right)} \right\} \quad or
$$
$$
T(s(X_{m+1}, y)) \leq A^T_{\left(k+b_-(k,\beta)\right)},
$$
(9)

*where $k := r_{m+1,m}(T(s(X_{test}, y)))$. Then, we have*

$$
L(\beta, m) \leq \mathbb{P}\Big(Y_{m+1} \in \widehat{C}_{RSA,T}(X_{m+1})\Big) \leq U(\beta, m).
$$

Note that any monotone transformation may be used in the above procedure, without affecting the finite-sample coverage bounds. We now describe a particular choice based on optimal transport (OT), which provides a canonical and data-adaptive alignment between real and reference score distributions.

#### 3.6.2. OPTIMAL TRANSPORT BASED SCORE ALIGNMENT

We introduce a score alignment mechanism based on one-dimensional optimal transport. This construction yields a continuous, rank-preserving transformation that adapts the scale of the real scores to that of the reference scores.

Let $\tilde{P} := \frac{1}{m} \sum_{i=1}^m \delta_{S_i}$, $\tilde{Q} := \frac{1}{N} \sum_{j=1}^N \delta_{\tilde{S}_j}$ denote the empirical distributions of the real calibration scores and the reference scores, respectively. In one dimension, the optimal transport coupling between $\tilde{P}$ and $\tilde{Q}$ is uniquely characterized by monotone rearrangement. When $m \neq N$, a real score $S_{(i)}$ may split its mass across adjacent reference scores under the optimal transport plan. Let $\tilde{P}_{ij}$ denote the corresponding OT coupling matrix (see Appendix C.6 for technical details on how to construct this matrix). We define the *OT barycentric alignment map* $T : \mathbb{R} \to \mathbb{R}$ by

$$
T(S_{(i)}) := \frac{\sum_j \tilde{P}_{ij} \tilde{S}_{(j)}}{\sum_j \tilde{P}_{ij}}.
$$
(10)

This map corresponds to the barycentric projection of the optimal transport plan and defines a continuous, monotone

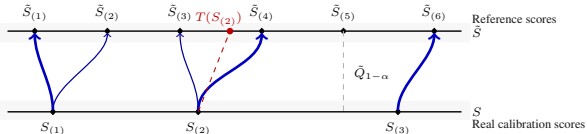

*Figure 2.* **1D OT coupling and barycentric alignment.** Mass is transported monotonically from real scores (bottom) to reference scores (top). Arrow thickness indicates transported mass; when $m \neq N$, a score may split across adjacent reference points. The barycentric projection defines a rank-preserving alignment.

transformation from real to reference score space. We call prediction sets constructed using this alignment map in $\widehat{C}_{RSA,T}(X_{m+1})$ from Proposition 3.3 as RSA-CP (OT) prediction sets (Figure 2 illustrates this concept.).

While score alignment via optimal transport is designed to reduce the distributional gap between real and reference scores, small calibration sample sizes $m$ fundamentally limit the strength of distribution-free coverage guarantees (cf. Section 1 and Appendix A). In particular, even after alignment, one cannot in general recover exact finite-sample coverage without additional assumptions. Nevertheless, if the alignment map substantially closes the distributional gap between the real and reference score distributions, it is possible to derive informative distribution-robust coverage bounds. The following result quantifies this effect.

**Theorem 3.4** (Distribution-Robust Coverage Analysis). *Suppose the real calibration set $\{(X_i, Y_i)\}_{i=1}^{m}$ is exchangeable with the test point $(X_{m+1}, Y_{m+1})$. Let reference scores $\{\tilde{S}_1, \ldots, \tilde{S}_N\}$ drawn i.i.d. from the continuous distribution $Q$. Let $P_s$ denote the real score distribution and define the transformed score law $T_{\#}P_s$ induced by a continuous, monotone mapping $T : \mathbb{R} \to \mathbb{R}$. Let $(T_{\#}P_s)_{m+1}^{(r)}$ and $Q_{m+1}^{(r)}$ denote the distributions of the $r$-th order statistic among $m+1$ i.i.d. draws from $T_{\#}P_s$ and $Q$, respectively. Then the prediction set $\widehat{C}_{RSA,T}(X_{m+1})$ in the Proposition 3.3 satisfies*

$$1 - \alpha - \varepsilon_{m,N}^{(T)}(P_S, Q) \leq \mathbb{P}\left\{ Y_{m+1} \in \widehat{C}(X_{m+1}) \right\}$$
$$\leq 1 - \alpha + \varepsilon_{m,N}^{(T)}(P_S, Q) + \frac{1}{N + m + 1},$$

*where $\varepsilon_{m,N}^{(T)}(P_S, Q)$ is as defined in (18).*

Thus, whenever the distributional gap is small, bounds in Theorem 3.4 are tighter around the target coverage than in Proposition 3.3. But when it is not, we still have finite-sample bounds in Proposition 3.3 which are computable.

*Remark* 3.5 (When are efficiency gains expected?). The main limitation of SCP in small samples is coarse rank resolution: with $m$ calibration samples, conformal ranks lie on a grid of size $(m+1)^{-1}$, often leading to conservative threshold estimates and wide prediction sets. RSA-CP refines this

rank structure by augmenting the calibration scores with $N$ reference scores, effectively improving the rank resolution to $(m + N + 1)^{-1}$. Reference scores are incorporated selectively through Beta-Binomial high-probability rank windows. When the reference and real score distributions are well aligned, reference scores frequently fall within these admissible windows, refining the rank structure and reducing conservatism. Under severe mismatch, incompatible reference scores are rarely included, and RSA-CP automatically reverts toward SCP-like behavior. Proposition 3.3 guarantees finite-sample validity regardless of the reference distribution, while Theorem 3.4 shows that the efficiency gains depend on the discrepancy between the transformed real score distribution and the reference distribution. Consequently, OT alignment plays a key role in moving RSA-CP toward the oracle augmented regime, where near-maximal efficiency gains can be achieved. Increasing $N$ refines the effective rank grid and stabilizes empirical reference quantiles, which can improve alignment and yield tighter prediction sets when the reference scores are informative. Under mismatch, however, incompatible reference scores are naturally filtered by the inclusion rule, preventing spurious efficiency gains and preserving validity. As a result, the gains from increasing $N$ eventually saturate once the rank grid becomes sufficiently fine. Overall, RSA-CP trades exact calibration at level $(1 - \alpha)$ for improved efficiency, while retaining distribution-free guarantees and recovering near-target coverage under good alignment. For practical guidance on choosing the reference score distribution, see Appendix E.

## 4. Simulation Study

We study the finite-sample behavior of RSA-CP in regimes where standard conformal calibration is known to be unstable. The simulations are designed to isolate three key effects: (i) rank coarseness under small calibration sizes, (ii) stabilization as the reference score resolution increases, and (iii) robustness to mismatch between real and reference score distributions.

We consider nonlinear regression models $Y = f(X) + \varepsilon$ with $X \in \mathbb{R}^5$ having heterogeneous, non-Gaussian marginals, yielding irregular and heavy-tailed score distributions. Two noise regimes are studied: skewed (log-normal) and heavy-tailed ($t_3$). Reference scores are generated from a deliberately misspecified generator with a *rare-shock* component (probability 0.05), designed to stress-test robustness as the reference size $N$ grows.

We compare SCP, SPI (Bashari et al., 2026) and their Synthetic-only method, RSA-CP (OT), and a reference score-only baseline. All experiments use $m = 20$, $\alpha = 0.05$, and $\beta = 0.4$, with $N \in \{20, 50, \ldots, 3000\}$. We report empirical coverage, average prediction-set size, and runtime over 100 trials in Figure 3. Full data-generation

and generator details are provided in Appendix F.1. For these simulation settings, the distribution-free finite-sample bounds from Proposition 3.3 yield $0.9048 \leq \mathbb{P}\{Y_{m+1} \in \widehat{C}(X_{m+1})\} \leq 1$, uniformly over all considered augmentation sizes $N \in \{20, 50, \ldots, 3000\}$. Our simulation results show that the empirical coverage consistently remains within this certified range across all values of $N$.

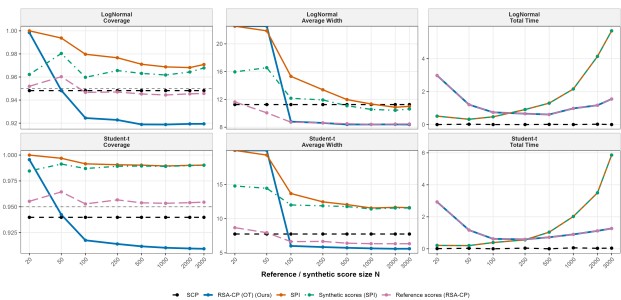

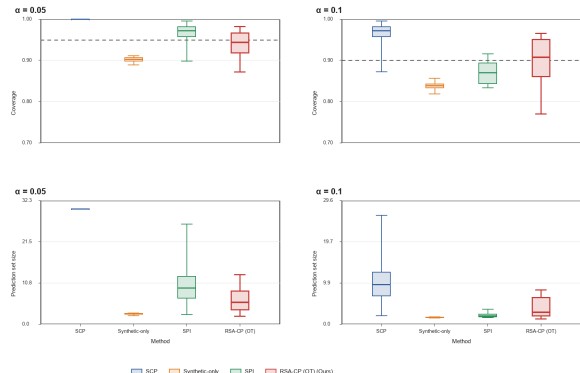

*Figure 4.* Image Classification: Coverage levels and widths of prediction sets constructed using SCP, RSA-CP(OT), SPI, and Synthetic data only methods for $\alpha = 0.05, 0.10$, augmented score size 1000, calibration size 15.

*Figure 3.* Simulation results as a function of reference score size ($N$). (A) empirical coverage, (B) prediction-set width, and (C) total runtime.

Figure 3 shows that SCP severely overcovers and produces wide, unstable prediction sets due to coarse empirical quantiles at $m = 20$. SPI is sensitive to generator misspecification, particularly under skewed and heavy-tailed noise. In contrast, RSA-CP exhibits monotonic efficiency gains and rapidly stabilizes coverage as augmentation score size increases, even under imperfect reference generators. These trends align with our theory: increasing reference size refines the effective rank grid, while OT alignment mitigates distributional mismatch.

RSA-CP also offers substantial computational advantages over SPI, avoiding repeated feature-based simulation and inference. Notably, even under heavy-tailed noise where the Gamma reference is misspecified, RSA-CP maintains valid coverage and achieves the tightest intervals among all methods. Additional sensitivity analyses, including varying calibration sizes and reference distributions, are reported in Appendix F.1.

## 5. Real Data Analysis

We evaluate RSA-CP on an ImageNet-derived image classification benchmark introduced in the SPI framework (Bashari et al., 2026). We follow the *same* dataset construction and experimental protocol as SPI to enable a controlled comparison, varying only the calibration/augmentation strategy while keeping the underlying classifier, features, and preprocessing fixed. We compare SCP, SPI (Bashari et al., 2026), their Synthetic-only baseline, and our RSA-CP (OT).

Figure 4 reports empirical coverage and prediction-set sizes for target miscoverage $\alpha \in \{0.05, 0.10\}$ with $n_{\text{cal}} = 15$, $n_{\text{test}} = 15000$, and $n_{\text{syn}} = 1000$ (100 trials). In this ex-

tremely small calibration regime, SCP becomes overly conservative and frequently returns vacuous prediction sets (often close to the full label space), yielding non-informative outputs. Augmentation-based baselines (SPI and synthetic-only) produce smaller sets, but at the cost of noticeable under-coverage in this setting. RSA-CP(OT) achieves a more favorable trade-off: it substantially reduces set sizes *while remaining closer to the target coverage*, delivering informative prediction sets without requiring a learned synthetic generator. This highlights a key practical advantage of RSA-CP: improved resolution and stability via cheap reference-score sampling and alignment, rather than resource-intensive synthetic data generation.

We further examine robustness with respect to calibration size and augmentation size. Figure 5 shows that as $n_{\text{cal}}$ increases, RSA-CP(OT) tracks the target coverage more closely, whereas SPI and the synthetic-only baseline remain systematically below target in this experiment. In addition, increasing $n_{\text{syn}}$ improves stability for all augmentation-based methods (Figure 8), but RSA-CP(OT) stabilizes quickly (typically after a few hundred reference scores), consistent with the rank-resolution effect: once the augmented rank grid is sufficiently refined, additional reference scores yield diminishing returns. Overall, these results demonstrate that RSA-CP provides a simple, scalable mechanism for producing non-vacuous prediction sets with reliable coverage in real-world, small-calibration regimes.

## 6. Conclusion

We introduced RSA-CP, a rank-based framework for improving conformal prediction in low-calibration regimes by augmenting calibration with user-chosen reference scores. By shifting attention from numerical score values to rank resolution, our approach constructs informative prediction sets

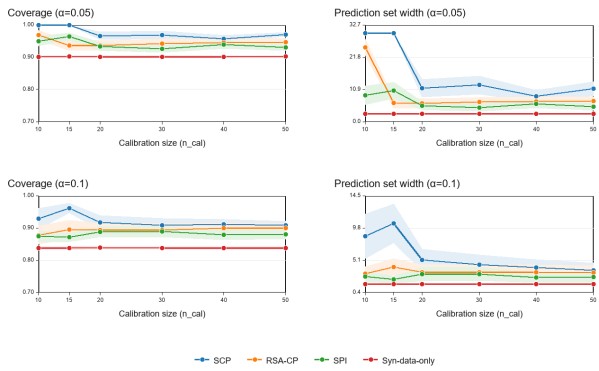

*Figure 5.* Image Classification: Coverage levels and widths of prediction sets constructed using SCP, RSA-CP(OT), SPI, and Synthetic data only methods for $\alpha = 0.05, 0.10$, augmented score size 1000 with increasing calibration sizes from 10 to 50.

while retaining explicit finite-sample coverage guarantees. The framework is modular, computationally lightweight, and compatible with arbitrary score functions and reference distributions.

Our analysis highlights conditional rank localization as the key mechanism underlying stability gains from augmentation, and shows how rank-preserving score alignment can further tighten coverage behavior when the reference distribution is well matched. In contrast to recent approaches that rely on training and deploying synthetic data generators to achieve finite-sample guarantees, RSA-CP attains similar stability gains through a direct, generator-free mechanism that requires only reference scores and rank-based computations.

Several directions remain open. Promising avenues include extending RSA-CP to conditional and fairness-aware conformal settings, incorporating prior or knowledge-based reference scores from a Bayesian perspective, developing adaptive reference selection strategies, and integrating the framework with conditional or group-conditional conformal guarantees.

## Acknowledgement

Bei Jiang and Linglong Kong were partially supported by grants from the Canada CIFAR AI Chairs program, the Alberta Machine Intelligence Institute (AMII), and Natural Sciences and Engineering Council of Canada (NSERC), and Linglong Kong was also partially supported by grants from the Canada Research Chair program from NSERC. The authors would also like to thank the anonymous reviewers for their constructive comments that improved the quality of this article.

## Impact Statement

The RSA-CP framework introduced in this work significantly advances the reliability of uncertainty quantification in automated decision-making systems. A key contribution of our approach is the ability to leverage high-resolution reference scores via rank-alignment operators to enhance limited real-world calibration sets, without requiring the dynamic generation of synthetic data (images, text, sound, etc.) during inference. By establishing a mathematically rigorous bridge between small-scale real data and large-scale synthetic distributions, RSA-CP provides valid, finite-sample coverage guarantees even in data-constrained regimes. This research fosters Trustworthy AI by enabling models to communicate their own uncertainty through reliable prediction sets, reducing the risks of overconfident predictions in high-stakes environments. Ultimately, this work offers a scalable and computationally efficient path toward more transparent, accountable, and robust AI systems across a broad range of predictive tasks. Nevertheless, the proposed RSA-CP framework is model-agnostic, and do not introduce new capabilities that would inherently raise ethical concerns beyond those already associated with predictive modeling and data-driven decision systems. We do not envision direct negative societal impacts stemming uniquely from this work, though, as with all uncertainty quantification methods, practitioners must take care to interpret and deploy prediction sets appropriately in context-sensitive applications.

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

# APPENDIX

## A. Related Work

**Conformal Prediction Efficiency.** The construction of distribution-free prediction sets dates back to foundational statistical theories (Wilks, 1941; 2012; Wald, 1943; Tukey, 1947; Scheffe & Tukey, 1945), with modern CP established by (Vovk et al., 2005; Saunders et al., 1999). Since its inception, CP has emerged as a versatile framework for uncertainty quantification across various domains, including regression (Lei et al., 2018), classification (Angelopoulos & Bates, 2021), and time-series analysis (Zaffran et al., 2022). While CP offers robust marginal coverage, its predictive efficiency, often measured by the set size, is fundamentally constrained by the calibration sample size $m$. This limitation is particularly acute in information retrieval and low-data regimes, where obtaining reliable confidence intervals is critical to prevent overconfident estimations under sparse feedback (Oosterhuis et al., 2024).

**Refined Coverage Guarantees.** Recent literature has focused on extending CP to offer guarantees beyond marginal coverage, such as group-conditional coverage (Vovk, 2015; 2012; Romano et al., 2019), local coverage (Guan, 2023), and label-conditioned guarantees (Sadinle et al., 2019). Despite these advancements, these methods often suffer from high variance or excessive interval widths when the specific group-of-interest lacks sufficient labeled data. Our work addresses this gap by augmenting limited real calibration information at the score/rank level using reference scores, without requiring generated feature-label samples.

**Prediction-Powered and Auxiliary Inference.** A parallel line of research, Prediction-Powered Inference (PPI) (Angelopoulos et al., 2023) and its stratified variants (Fisch et al., 2024), leverages large-scale unlabeled data to enhance statistical power. Related approaches explore using unlabeled data for efficiency (Tibshirani et al., 2019) or few-shot CP for auxiliary tasks (Park et al., 2020). However, these methods typically assume that unlabeled or auxiliary data are drawn from the same distribution $P$, or they rely on asymptotic validity. In contrast, we focus on finite-sample guarantees under unknown distributional shifts where $Q \neq P$.

**The SPI Framework and Score Transport.** The central mechanism of the Synthetic-Powered Predictive Inference (SPI) framework (Bashari et al., 2026) is the score transporter. This component is a data-driven function $T$, typically implemented as an empirical quantile mapping ($T_{\text{EQM}}$), whose primary role is to align the distribution of scores from the small, trusted real set with the abundant synthetic data. Specifically, $T_{\text{EQM}}$ preserves the rank order of real scores while correcting their magnitudes based on the synthetic reference scale. By transferring the stability properties of the large synthetic distribution onto the real scores, SPI allows for prediction sets that benefit from the synthetic domain's sample size while maintaining distribution-free coverage tied to the real data.

A key highlight of our approach is the ability to leverage high-resolution references scores via rank-alignment operators to enhance limited real-world calibration sets, without requiring the dynamic generation of synthetic data (images, text, sound, etc.) during inference. By establishing a mathematically rigorous bridge between small-scale real data and large-scale synthetic distributions, RSA-CP provides valid, finite-sample coverage guarantees even in data-constrained regimes. This research fosters Trustworthy AI by enabling models to communicate their own uncertainty through reliable prediction sets, reducing the risks of overconfident predictions in high-stakes environments.

## B. Algorithm

---

**Algorithm 2** Certified selection of $\beta$

---

1: **Input:** $m, N \in \mathbb{N}$, miscoverage $\alpha \in (0, 1)$, search grid $\mathcal{B} \subset (0, 1)$ (or tolerance $\varepsilon > 0$ for bisection).
2: **Output:** Certified $\beta_{\text{cert}}$ (if exists) and certified lower bound $L(\beta_{\text{cert}}, m)$.
3: Set cutoff index $j^\star \leftarrow \lceil (1 - \alpha)(m + N + 1) \rceil$.
4: Initialize $\beta_{\text{cert}} \leftarrow$ None and $L_{\max} \leftarrow 0$.
5: **for** each $\beta \in \mathcal{B}$ (sorted in decreasing order) **do**
6:     Compute $b_+(k, \beta)$ for all $k = 1, \ldots, m + 1$ and then compute $L(\beta, m)$.
7:     **if** $L(\beta, m) \geq 1 - \alpha$ **then**
8:         Set $\beta_{\text{cert}} \leftarrow \beta$ and **break**.
9:     **end if**
10:     **if** $L(\beta, m) > L_{\max}$ **then**
11:         $L_{\max} \leftarrow L(\beta, m)$.
12:     **end if**
13: **end for**
14: **if** $\beta_{\text{cert}} =$ None **then**
15:     **Return:** No certified $\beta$ exists on $(0, 1)$; optionally report $\beta^\dagger := \arg\max_{\beta \in \mathcal{B}} L(\beta, m)$ and $L_{\max}$.
16: **else**
17:     **Return:** $\beta_{\text{cert}}$ and $L(\beta_{\text{cert}}, m)$.
18: **end if**

---

## C. Technical details and Proofs

### C.1. The Beta-Binomial Distribution

The Beta-Binomial distribution arises naturally when modeling uncertainty over a Binomial success probability. It plays a central role in our analysis of rank uncertainty under score augmentation.

**Definition.** Let $p \sim \text{Beta}(a, c)$ and, conditional on $p$, let $B \mid p \sim \text{Binomial}(N, p)$. The marginal distribution of $B$ is called the *Beta-Binomial* distribution with parameters $(N, a, c)$, denoted $B \sim \text{BetaBin}(N, a, c)$, and has probability mass function

$$\mathbb{P}(B = b) = \binom{N}{b} \frac{\mathcal{B}(b + a, \ N - b + c)}{\mathcal{B}(a, c)}, \qquad b = 0, 1, \ldots, N, \tag{11}$$

where $\mathcal{B}(\cdot, \cdot)$ denotes the Beta function. The Beta function $\text{B}(a, b)$ appearing in (11) is defined as

$$\text{B}(a, c) := \int_0^1 t^{a-1}(1 - t)^{c-1} \, dt, \qquad a, c > 0. \tag{12}$$

Equivalently, it can be expressed in terms of the Gamma function as

$$\text{B}(a, c) = \frac{\Gamma(a)\Gamma(c)}{\Gamma(a + c)}. \tag{13}$$

The Beta-Binomial distribution can be viewed as a Binomial model with a random success probability.

In our setting, the Beta-Binomial distribution characterizes the conditional distribution of the number of reference scores falling below a test score, given its rank among the real calibration scores. Specifically, conditioning on a real-score rank $k$ induces a $\text{BetaBin}(N, k, m + 2 - k)$ law, which governs the uncertainty window $\left[ b_-(k, \beta), \ b_+(k, \beta) \right]$ used in the RSA-CP inclusion rule.

The lower and upper window indices $b_-(k, \beta)$ and $b_+(k, \beta)$ are defined as the $\beta/2$ and $1 - \beta/2$ quantiles of the corresponding Beta-Binomial distribution. These quantiles are monotone in $\beta$ and can be computed efficiently using standard numerical routines.

C.1.1. BETA-BINOMIAL WINDOWS:

*Proof.* Let $S_1, \ldots, S_m, S_{m+1}$ be i.i.d. from a continuous distribution, and let $r_{m+1,m}$ denote the rank of $S_{m+1}$ among $\{S_1, \ldots, S_m, S_{m+1}\}$ (with rank 1 corresponding to the smallest value). Since the joint law is exchangeable and ties occur with probability zero, all $(m+1)!$ strict orderings of $(S_1, \ldots, S_{m+1})$ are equally likely. For each $k \in \{1, \ldots, m+1\}$, exactly $m!$ of these orderings place $S_{m+1}$ in position $k$, hence

$$\mathbb{P}(r_{m+1,m} = k) = \frac{m!}{(m+1)!} = \frac{1}{m+1},$$

which proves (3).

Now augment the calibration set with $N$ additional i.i.d. scores $\tilde{S}_1, \ldots, \tilde{S}_N$, independent of $(S_i)_{i=1}^{m+1}$ and drawn from the same continuous distribution. Define

$$B := \sum_{j=1}^{N} \mathbf{1}\{\tilde{S}_j \leq S_{m+1}\}.$$

By continuity, $\mathbb{P}(\tilde{S}_j = S_{m+1}) = 0$, so

$$r_{m+1,m+N+1} = r_{m+1,m} + B.$$

Fix $k \in \{1, \ldots, m+1\}$ and condition on the event $\{r_{m+1,m} = k\}$. Write $F$ for the (continuous) CDF of the score distribution and set

$$U_i := F(S_i), \quad i = 1, \ldots, m+1, \qquad \tilde{U}_j := F(\tilde{S}_j), \quad j = 1, \ldots, N.$$

By the probability integral transform, $U_1, \ldots, U_{m+1}, \tilde{U}_1, \ldots, \tilde{U}_N$ are i.i.d. $\mathrm{Unif}(0,1)$. Moreover, ranks are preserved under monotone transforms, so the event $\{r_{m+1,m} = k\}$ is equivalent to "$U_{m+1}$ is the $k$-th order statistic among $U_1, \ldots, U_{m+1}$," i.e.

$$\{r_{m+1,m} = k\} \equiv \{U_{m+1} = U_{(k)}\},$$

where $U_{(k)}$ denotes the $k$-th order statistic of $m+1$ i.i.d. uniforms. It is standard that

$$U_{(k)} \sim \mathrm{Beta}(k, \, m+2-k),$$

with density

$$f_{U_{(k)}}(u) = \frac{1}{\mathcal{B}(k, m+2-k)} u^{k-1}(1-u)^{m+1-k}, \qquad u \in (0,1).$$

Next, observe that

$$B = \sum_{j=1}^{N} \mathbf{1}\{\tilde{S}_j \leq S_{m+1}\} = \sum_{j=1}^{N} \mathbf{1}\{\tilde{U}_j \leq U_{m+1}\}.$$

Conditionally on $U_{m+1} = u$, the indicators $\mathbf{1}\{\tilde{U}_j \leq u\}$ are i.i.d. $\mathrm{Bernoulli}(u)$, hence

$$(B \mid U_{m+1} = u) \sim \mathrm{Binomial}(N, u).$$

Therefore, conditioning on $\{r_{m+1,m} = k\}$ (equivalently $U_{m+1} = U_{(k)}$) yields a Beta-Binomial mixture:

$$\begin{aligned}
\mathbb{P}(B = b \mid r_{m+1,m} = k) &= \int_0^1 \mathbb{P}(B = b \mid U_{m+1} = u) \, f_{U_{(k)}}(u) \, du \\
&= \int_0^1 \binom{N}{b} u^b (1-u)^{N-b} \frac{u^{k-1}(1-u)^{m+1-k}}{\mathcal{B}(k, m+2-k)} \, du \\
&= \binom{N}{b} \frac{1}{\mathcal{B}(k, m+2-k)} \int_0^1 u^{b+k-1}(1-u)^{N-b+m+1-k} \, du \\
&= \binom{N}{b} \frac{\mathcal{B}(b+k, \, N-b+m+2-k)}{\mathcal{B}(k, \, m+2-k)},
\end{aligned}$$

for $b = 0, 1, \ldots, N$, which is the claimed Beta-Binomial pmf.

Finally, since $r_{m+1,m+N+1} = k + B$, for any $t \in \{k, k+1, \ldots, k+N\}$ we have

$$
\begin{aligned}
\mathbb{P}(r_{m+1,m+N+1} = t \mid r_{m+1,m} = k) &= \mathbb{P}(B = t - k \mid r_{m+1,m} = k) \\
&= \binom{N}{t-k} \frac{\mathcal{B}(t, \, m+N+2-t)}{\mathcal{B}(k, \, m+2-k)},
\end{aligned}
$$

which is exactly (4). $\qquad\square$

**Unconditional rank distribution.** Marginalizing over $k$ yields

$$
\mathbb{P}(r_{m+1,m+N+1} = t) = \sum_{k=1}^{m+1} \mathbb{P}(r_{m+1,m+N+1} = t \mid r_{m+1,m} = k) \, \mathbb{P}(r_{m+1,m} = k).
$$

Since the augmented scores, real scores, and test score are exchangeable under the present assumption, the rank is uniformly distributed:

$$
\mathbb{P}(r_{m+1,m+N+1} = t) = \frac{1}{m+N+1}, \qquad t = 1, \ldots, m+N+1.
$$

### C.2. Proof of Theorem 3.1

*Proof.* Let $k := r_{m+1,m}$ denote the rank of the test score among the $m$ real calibration scores (including the test score). By exchangeability of $(S_1, \ldots, S_m, S_{m+1})$ and continuity (no ties),

$$
\mathbb{P}(k = t) = \frac{1}{m+1}, \qquad t = 1, \ldots, m+1. \tag{14}
$$

For each realized $k$, define the (deterministic) inclusion indicator induced by (5):

$$
I_k := \mathbf{1}\{k + b_+(k, \beta) \le j^\star \ \text{ or } \ k + b_-(k, \beta) \le j^\star \le k + b_+(k, \beta)\}.
$$

Then, by construction of $\widehat{C}_{\mathrm{RSA}}$, we have

$$
\mathbf{1}\Big\{Y_{m+1} \in \widehat{C}_{\mathrm{RSA}}(X_{m+1})\Big\} = I_k, \qquad \text{almost surely.} \tag{15}
$$

We now sandwich $I_k$ between two simpler indicators. First, if $k + b_+(k, \beta) \le j^\star$, then the first clause in the union holds, hence $I_k = 1$. Therefore,

$$
\mathbf{1}\{k + b_+(k, \beta) \le j^\star\} \ \le \ I_k. \tag{16}
$$

Second, if $I_k = 1$, then either $k + b_+(k, \beta) \le j^\star$ or $k + b_-(k, \beta) \le j^\star \le k + b_+(k, \beta)$; in both cases it must hold that $k + b_-(k, \beta) \le j^\star$. Hence,

$$
I_k \ \le \ \mathbf{1}\{k + b_-(k, \beta) \le j^\star\}. \tag{17}
$$

Taking expectations in (16)–(17) and using (14) yields

$$
\mathbb{E}[\mathbf{1}\{k + b_+(k, \beta) \le j^\star\}] \le \mathbb{E}[I_k] \le \mathbb{E}[\mathbf{1}\{k + b_-(k, \beta) \le j^\star\}],
$$

$$
\frac{1}{m+1} \sum_{k=1}^{m+1} \mathbf{1}\{k + b_+(k, \beta) \le j^\star\} \le \mathbb{P}\Big\{Y_{m+1} \in \widehat{C}_{\mathrm{RSA}}(X_{m+1})\Big\} \le \frac{1}{m+1} \sum_{k=1}^{m+1} \mathbf{1}\{k + b_-(k, \beta) \le j^\star\}.
$$

By the definitions of $L(\beta, m)$ and $U(\beta, m)$, this is exactly the claimed bound. $\qquad\square$

### C.3. Proof of Proposition 3.2

*Proof.* By construction, (7) is obtained from (5) by applying the (generalized) inverse quantile map $\tilde{Q}^{-1}$ to the same rank thresholds. Since $\tilde{Q}^{-1}$ is nondecreasing, it preserves the order of thresholds, and ranks under $\tilde{Q}$ correspond to its empirical quantiles. Therefore, for every candidate label $y$, the mapped rule declares $y \in \widehat{C}(X_{m+1})$ by inclusion rule (7) if and only if the original rank-based rule (5) declares $y \in \widehat{C}(X_{m+1})$, implying they are equal almost surely. The stated bounds then follow immediately. $\qquad\square$

### C.4. Proof of Theorem 3.4

Let $Z_i := T(S_i)$, $i = 1, \ldots, m+1$, and let $R_{m+1} = 1 + \sum_{i=1}^{m} \mathbf{1}\{Z_i < Z_{m+1}\}$ denote the rank of the transformed test score $Z_{m+1}$ among $\{Z_1, \ldots, Z_m, Z_{m+1}\}$. By exchangeability of $\{Z_i\}_{i=1}^{m+1}$, we have $R_{m+1} \sim \mathrm{Unif}\{1, \ldots, m+1\}$.

Let $j^\star = \lceil (1-\alpha)(m+N+1) \rceil$. For each $r \in \{1, \ldots, m+1\}$, define the augmented calibration collection obtained when the test score occupies real-score rank $r$:

$$\mathcal{A}^{(-r)} = \{Z_{(1)}, \ldots, Z_{(r-1)}, Z_{(r+1)}, \ldots, Z_{(m+1)}\} \cup \{\widetilde{S}_1, \ldots, \widetilde{S}_N\},$$

where $Z_{(1)} \leq \cdots \leq Z_{(m+1)}$ are the order statistics of $Z_1, \ldots, Z_{m+1}$. Let $A_{(1)}^{(-r)} \leq \cdots \leq A_{(m+N)}^{(-r)}$ denote the ordered elements of $\mathcal{A}^{(-r)}$, with the convention $A_{(0)}^{(-r)} = -\infty$, $A_{(m+N+1)}^{(-r)} = +\infty$. Then the augmented conformal inclusion event for the true label can be written as

$$\{Y_{m+1} \in \widehat{C}(X_{m+1})\} = \{Z_{m+1} \leq A_{(j^\star)}^{(-R_{m+1})}\}.$$

Therefore, conditioning on the real-score rank $R_{m+1}$, we obtain

$$\mathbb{P}\Big\{Y_{m+1} \in \widehat{C}(X_{m+1})\Big\} = \sum_{r=1}^{m+1} \mathbb{P}\Big(Z_{m+1} \leq A_{(j^\star)}^{(-r)}, \ R_{m+1} = r\Big)$$

$$= \frac{1}{m+1} \sum_{r=1}^{m+1} \mathbb{P}_{P_T, Q}\Big\{Z_{(r)} \leq A_{(j^\star)}^{(-r)}\Big\},$$

where $P_T := T_\# P_S$ denotes the transformed real-score distribution, and the notation $\mathbb{P}_{P_T, Q}$ indicates that $Z_1, \ldots, Z_{m+1} \overset{\text{i.i.d.}}{\sim} P_T$ and $\widetilde{S}_1, \ldots, \widetilde{S}_N \overset{\text{i.i.d.}}{\sim} Q$, independently.

Now define the corresponding oracle probability $p_r^Q := \mathbb{P}_Q\Big\{W_{(r)} \leq B_{(j^\star)}^{(-r)}\Big\}$, where $W_1, \ldots, W_{m+1}, \widetilde{S}_1, \ldots, \widetilde{S}_N \overset{\text{i.i.d.}}{\sim} Q$, and $B_{(1)}^{(-r)} \leq \cdots \leq B_{(m+N)}^{(-r)}$ are the order statistics of

$$\{W_{(1)}, \ldots, W_{(r-1)}, W_{(r+1)}, \ldots, W_{(m+1)}\} \cup \{\widetilde{S}_1, \ldots, \widetilde{S}_N\}.$$

Since the event $\Big\{Z_{(r)} \leq A_{(j^\star)}^{(-r)}\Big\}$ is measurable with respect to the augmented ordered score collection, the total-variation inequality gives

$$\mathbb{P}_{P_T, Q}\Big\{Z_{(r)} \leq A_{(j^\star)}^{(-r)}\Big\} \geq p_r^Q - d_{\mathrm{TV}}\Big(\mathcal{L}_{P_T, Q}^{(r)}, \mathcal{L}_{Q, Q}^{(r)}\Big),$$

where $\mathcal{L}_{P_T, Q}^{(r)}$ denotes the joint law of $\Big(Z_{(r)}, A_{(j^\star)}^{(-r)}\Big)$ under $Z_1, \ldots, Z_{m+1} \overset{\text{i.i.d.}}{\sim} P_T$ and $\widetilde{S}_1, \ldots, \widetilde{S}_N \overset{\text{i.i.d.}}{\sim} Q$, while $\mathcal{L}_{Q, Q}^{(r)}$ denotes the corresponding oracle law under all $m+N+1$ scores drawn i.i.d. from $Q$.

Define the augmented-ordering discrepancy

$$\varepsilon_{m,N}^{(T)}(P_S, Q) := \frac{1}{m+1} \sum_{r=1}^{m+1} d_{\mathrm{TV}}\Big(\mathcal{L}_{P_T, Q}^{(r)}, \mathcal{L}_{Q, Q}^{(r)}\Big). \tag{18}$$

Then

$$\mathbb{P}\Big\{Y_{m+1} \in \widehat{C}(X_{m+1})\Big\} \geq \frac{1}{m+1} \sum_{r=1}^{m+1} p_r^Q - \varepsilon_{m,N}^{(T)}(P_S, Q).$$

Under the oracle law, all scores $W_1, \ldots, W_{m+1}, \widetilde{S}_1, \ldots, \widetilde{S}_N$ are i.i.d. from $Q$. Hence the rank of the test score among the $m+N+1$ augmented scores is uniform on $\{1, \ldots, m+N+1\}$. Therefore, by the standard conformal rank argument,

$$\frac{1}{m+1} \sum_{r=1}^{m+1} p_r^Q = \mathbb{P}_Q\{\mathrm{rank}(W_{m+1}) \leq j^\star\} \geq 1-\alpha.$$

Consequently,

$$\mathbb{P}\left\{Y_{m+1} \in \widehat{C}(X_{m+1})\right\} \geq 1 - \alpha - \varepsilon_{m,N}^{(T)}(P_S, Q).$$

The upper bound follows analogously. Applying the reverse total-variation inequality gives

$$\mathbb{P}_{P_T,Q}\left\{Z_{(r)} \leq A_{(j^\star)}^{(-r)}\right\} \leq p_r^Q + d_{\text{TV}}\left(\mathcal{L}_{P_T,Q}^{(r)}, \mathcal{L}_{Q,Q}^{(r)}\right).$$

Averaging over $r$ yields

$$\mathbb{P}\left\{Y_{m+1} \in \widehat{C}(X_{m+1})\right\} \leq \frac{1}{m+1}\sum_{r=1}^{m+1} p_r^Q + \varepsilon_{m,N}^{(T)}(P_S, Q).$$

Again using the oracle conformal rank argument,

$$\frac{1}{m+1}\sum_{r=1}^{m+1} p_r^Q \leq 1 - \alpha + \frac{1}{m+N+1}.$$

Therefore,

$$\mathbb{P}\left\{Y_{m+1} \in \widehat{C}(X_{m+1})\right\} \leq 1 - \alpha + \frac{1}{m+N+1} + \varepsilon_{m,N}^{(T)}(P_S, Q).$$

Combining the lower and upper bounds gives

$$1 - \alpha - \varepsilon_{m,N}^{(T)}(P_S, Q) \leq \mathbb{P}\left\{Y_{m+1} \in \widehat{C}(X_{m+1})\right\} \leq 1 - \alpha + \frac{1}{m+N+1} + \varepsilon_{m,N}^{(T)}(P_S, Q),$$

which proves the claim.

### C.5. Additional results

**Lemma C.1** (Monotonicity of the certified upper bound). *Fix $m, N \in \mathbb{N}$ and $\alpha \in (0,1)$, and let $j^\star = \lceil (1-\alpha)(N+1) \rceil$. Define*

$$U(\beta, m) = \frac{1}{m+1}\sum_{k=1}^{m+1} \mathbf{1}\{b_-(k, \beta) \leq j^\star\},$$

*where $b_-(k, \beta) = Q_{B|k}(\beta/2)$ and $B \mid k \sim \text{BetaBin}(N, k, m+2-k)$. Then $U(\beta)$ is nonincreasing in $\beta$ on $(0,1)$.*

*Proof.* Fix $k$. The quantile function $Q_{B|k}(p)$ is nondecreasing in $p$. If $\beta_1 < \beta_2$, then $\beta_1/2 < \beta_2/2$, hence

$$b_-(k, \beta_1) = Q_{B|k}(\beta_1/2) \leq Q_{B|k}(\beta_2/2) = b_-(k, \beta_2).$$

Therefore, the indicator $\mathbf{1}\{b_-(k, \beta) \leq j^\star\}$ is nonincreasing in $\beta$ for each fixed $k$ (as the threshold $b_-(k, \beta)$ moves upward with $\beta$). Summing over $k$ and dividing by $m+1$ yields that $U(\beta, m)$ is nonincreasing in $\beta$. □

**Lemma C.2** (Monotonicity of the lower bound). *Fix $m, N \in \mathbb{N}$ and $\alpha \in (0,1)$, and let $j^\star = \lceil (1-\alpha)(N+1) \rceil$. Define*

$$L(\beta, m) = \frac{1}{m+1}\sum_{k=1}^{m+1} \mathbf{1}\{b_+(k, \beta) \leq j^\star\}$$

*where $b_+(k, \beta) = Q_{B|k}(1 - \beta/2)$ and $B \mid k \sim \text{BetaBin}(N, k, m+2-k)$. Then $L(\beta, m)$ is nondecreasing in $\beta$ on $(0,1)$.*

*Proof.* Fix $k \in \{1, \ldots, m+1\}$. The Beta-Binomial quantile function is nondecreasing in its argument, i.e., $Q_{B|k}(p_1) \leq Q_{B|k}(p_2)$ whenever $p_1 \leq p_2$. If $\beta_1 < \beta_2$, then $1 - \beta_1/2 > 1 - \beta_2/2$, hence

$$b_+(k, \beta_1) = Q_{B|k}(1 - \beta_1/2) \geq Q_{B|k}(1 - \beta_2/2) = b_+(k, \beta_2).$$

Therefore, the indicator $\mathbf{1}\{b_+(k, \beta) \leq j^\star\}$ is nondecreasing in $\beta$ for each fixed $k$. Summing over $k$ and dividing by $m+1$ yields that $L(\beta, m)$ is nondecreasing in $\beta$. □

## C.6. Computing the 1D Optimal Transport Coupling

We describe how to compute the optimal transport (OT) coupling matrix $\tilde{P}_{ij}$ between the empirical measures

$$\tilde{P} := \frac{1}{m} \sum_{i=1}^{m} \delta_{S_i}, \qquad \tilde{Q} := \frac{1}{N} \sum_{j=1}^{N} \delta_{\tilde{S}_j},$$

in one dimension under the cost $c(x, y) = |x - y|^p$ for any $p \geq 1$. In 1D, the optimal coupling is given by monotone rearrangement and can be computed by matching cumulative masses of the sorted atoms. When $m \neq N$, a source atom may split its mass across at most two consecutive target atoms.

Let $S_{(1)} \leq \cdots \leq S_{(m)}$ and $\tilde{S}_{(1)} \leq \cdots \leq \tilde{S}_{(N)}$ be the sorted real and reference scores. Each source atom has mass $a_i := 1/m$ and each target atom has mass $b_j := 1/N$.

Initialize pointers $i \leftarrow 1$, $j \leftarrow 1$ and remaining masses $r \leftarrow a_1$, $c \leftarrow b_1$. Set $\tilde{P}_{ij} = 0$ for all $i, j$. Iterate:

1. Move mass $\Delta := \min\{r, c\}$ from source $i$ to target $j$ by setting $\tilde{P}_{ij} \leftarrow \tilde{P}_{ij} + \Delta$.

2. Update remainders $r \leftarrow r - \Delta$ and $c \leftarrow c - \Delta$.

3. If $r = 0$, increment $i \leftarrow i + 1$ and set $r \leftarrow a_i$ (if $i \leq m$).

4. If $c = 0$, increment $j \leftarrow j + 1$ and set $c \leftarrow b_j$ (if $j \leq N$).

Stop when $i > m$ or $j > N$. The resulting matrix $\tilde{P}$ is a valid coupling, and in 1D it is optimal for any convex cost $|x - y|^p$.

**Barycentric alignment map.** Given the coupling, the OT barycentric alignment map is computed on the sorted real scores as

$$T(S_{(i)}) := \frac{\sum_{j=1}^{N} \tilde{P}_{ij} \tilde{S}_{(j)}}{\sum_{j=1}^{N} \tilde{P}_{ij}} = m \sum_{j=1}^{N} \tilde{P}_{ij} \tilde{S}_{(j)}, \tag{19}$$

since $\sum_j \tilde{P}_{ij} = 1/m$ for each $i$. This defines a monotone, piecewise-linear transport map from the real score space to the reference score space.

# D. Asymptotic Regimes of RSA-CP

In this section, we provide discussion of the asymptotic behavior of RSA-CP under different scaling regimes for the calibration sample size $m$ and the number of reference scores $N$.

## D.1. Regime 1: Fixed Number of Reference Scores

Consider the regime in which the calibration sample size satisfies $m \to \infty$ while the number of reference scores $N$ remains fixed. In this setting, SCP already achieves quantile resolution of order $(m + 1)^{-1}$, and the contribution of the $N$ reference scores becomes asymptotically negligible relative to the calibration sample.

Consequently, the augmented rank grid $(m + N + 1)^{-1}$ is asymptotically equivalent to the SCP rank grid $(m + 1)^{-1}$. Hence, RSA-CP does not provide efficiency improvements over SCP in this regime. This reflects the fact that the main benefits of RSA-CP arise in small-sample settings where coarse rank resolution is a dominant source of conservatism.

## D.2. Regime 2: Proportional Growth of Calibration and Reference Scores

Now consider the regime in which both $m$ and $N$ grow proportionally, with $N/m \to c \in (0, \infty)$. Assume additionally that the OT alignment map converges to a population transport map $T^\star$ satisfying $T^\star_\# P = Q$, where $T^\star_\#$ denotes the pushforward measure.

In this regime, the total variation discrepancy term appearing in Theorem 3.4 vanishes asymptotically. Consequently, RSA-CP (OT) attains exact asymptotic coverage:

$$\lim_{m,N \to \infty} \mathbb{P}\{Y_{n+1} \in \widehat{C}_{n,\alpha}^{\text{RSA-CP(OT)}}(X_{n+1})\} = 1 - \alpha.$$

Moreover, since the inclusion rule is rank-preserving and the OT alignment becomes asymptotically exact, the resulting prediction sets become asymptotically equivalent to those obtained from the oracle augmented procedure using additional i.i.d. draws from the true score distribution. Thus, RSA-CP asymptotically recovers the oracle regime under sufficiently accurate alignment.

## E. Practical Guidance for Choosing Reference Distributions

Proposition 3.3 and Theorem 3.4 should be interpreted together as a practical guide for selecting reference score distributions. The key insight is that the choice of reference distribution affects *efficiency* but not *validity*. In particular, the distribution-free coverage guarantees of Proposition 3.3 hold for *any* choice of reference distribution. Consequently, even a poorly chosen reference distribution cannot invalidate the finite-sample coverage guarantee. The role of Theorem 3.4 is complementary: it quantifies how tightly the achieved coverage concentrates around the target level $1 - \alpha$. This tightening improves as the discrepancy between the transported real-score distribution and the reference distribution decreases.

In practice, we recommend selecting a simple parametric family that matches the structural properties of the score distribution. For example:

- Beta distributions are natural for scores supported on bounded intervals such as $[0, 1]$;

- Log-normal or Gamma distributions are appropriate for strictly positive and potentially heavy-tailed scores;

- Gaussian approximations may be suitable when the score distribution is approximately symmetric.

The parameters of the reference distribution can then be estimated from the observed calibration scores using standard procedures such as maximum likelihood estimation or moment matching. In our real-data experiments, we used simple parametric approximations estimated from the observed calibration scores and found them to perform well in practice.

The subsequent optimal transport (OT) alignment step plays an important corrective role. Even when the chosen parametric family is only an approximation to the true score distribution, the OT transport map partially compensates for the mismatch by aligning the reference-score distribution with the empirical calibration-score distribution. Thus, the method does not rely on exact parametric specification.

When additional information is available, more informative reference distributions may be constructed from auxiliary sources. Examples include:

- historical datasets from related prediction tasks,

- external control cohorts in clinical studies (U.S. Food and Drug Administration, 2023),

- population-level score distributions from prior deployments,

- synthetic scores generated from large pretrained models (Bashari et al., 2026).

Such auxiliary information can substantially improve alignment and thereby improve efficiency. Nevertheless, Proposition 3.3 continues to provide a finite-sample safety guarantee regardless of the quality of the chosen reference distribution. This robustness is important in practice, since the method remains valid even when the reference distribution is misspecified or only weakly related to the target score distribution.

**Example: Small-Sample Clinical Trials**    Small-sample clinical trials, such as rare disease studies or early-phase clinical investigations, often suffer from insufficient calibration data, causing standard split conformal prediction (SCP) to become vacuous or overly conservative due to coarse rank resolution. RSA-CP directly addresses this limitation by leveraging external information, such as historical trials, patient registries, external control arms, or real-world evidence, to construct reference score distributions, an approach increasingly supported by recent regulatory guidance (e.g., U.S. FDA (U.S. Food and Drug Administration, 2023)). Unlike existing borrowing strategies (Zhu et al., 2025; Gao et al., 2025b), which typically operate at the data, likelihood, or model level, RSA-CP borrows *rank resolution*. The reference scores refine the effective quantile resolution available in small samples, enabling less conservative conformal thresholds while preserving finite-sample validity. Importantly, the validity guarantees of Proposition 3.3 hold regardless of the quality or alignment

of the external information. Theorem 3.4 further quantifies how efficiency improves as the transported reference-score distribution becomes better aligned with the target score distribution. This provides a principled mechanism for incorporating external information without requiring strict exchangeability between historical and current populations, making RSA-CP particularly well suited for settings where auxiliary data are informative but not perfectly matched to the target clinical population.

# F. Additional Experiments

## F.1. Simulation study

We conduct simulation studies to examine the finite-sample behavior of prediction sets constructed using rank-based score augmentation and alignment. The experiments are designed to reflect the core phenomena studied in this paper: instability under small calibration sizes, the effect of increasing reference score resolution, and robustness to mismatch between real and reference score distributions.

### F.1.1. DATA-GENERATING PROCESSES

We generate i.i.d. covariates $X_i \in \mathbb{R}^5$ with heterogeneous, non-Gaussian marginals to induce realistic and irregular score distributions. Specifically, $X_{i1}$ follows a two-component Gaussian mixture, $X_{i2} \sim t_3$ (scaled), $X_{i3}$ is Log-Normal, $X_{i4}$ follows a shifted Beta distribution, and $X_{i5} \sim \mathrm{Unif}(0,1)$. The response is generated as

$$Y_i = f(X_i) + \varepsilon_i,$$

where $f$ is a nonlinear regression function and $\varepsilon_i$ is an additive noise term. We consider two noise regimes:

- **Skewed errors.** $\varepsilon_i \sim \mathrm{LogNormal}(0, 0.5^2)$, yielding highly asymmetric score distributions.

- **Heavy-tailed errors.** $\varepsilon_i \sim t_3$, producing frequent large deviations and extreme scores.

These settings stress-test calibration procedures under skewness and heavy tails, both known to cause instability when calibration sizes are small.

### F.1.2. REFERENCE SCORES AND GENERATOR MISSPECIFICATION

A central goal of our simulations is to assess how increasing the number of reference scores stabilizes rank-based calibration, even when the reference distribution is imperfect. Rather than assuming access to an oracle sampler, we deliberately use *misspecified reference score generators*.

Reference scores are generated from an approximate residual simulator that captures the bulk behavior of the true score distribution but is noisy and imperfect. To model realistic generator failures, we introduce a *rare shock* mechanism: with probability $p = 0.05$, the generator outputs an extreme outlier far from the main distribution. When the reference sample size $N$ is small, these shocks induce substantial variability in the empirical reference quantiles; as $N$ grows, their influence is mitigated. This setup directly probes the stability benefits predicted by our rank-resolution analysis.

### F.1.3. METHODS COMPARED

We compare the following prediction set constructions:

- **SCP.** Standard split conformal prediction using only the real calibration scores.

- **SPI: Feature-based synthetic augmentation.** Recently developed method that augments calibration using feature-conditioned synthetic samples, which act as another baseline.

- **RSA-CP (OT) Score-aligned prediction set (ours).** The prediction set $\widehat{C}_{RSA,T}(X_{m+1})$ from Proposition 3.3 with OT alignment map.

- **Reference score-only baseline.** Calibration performed using reference scores alone, included to highlight bias under misspecification.

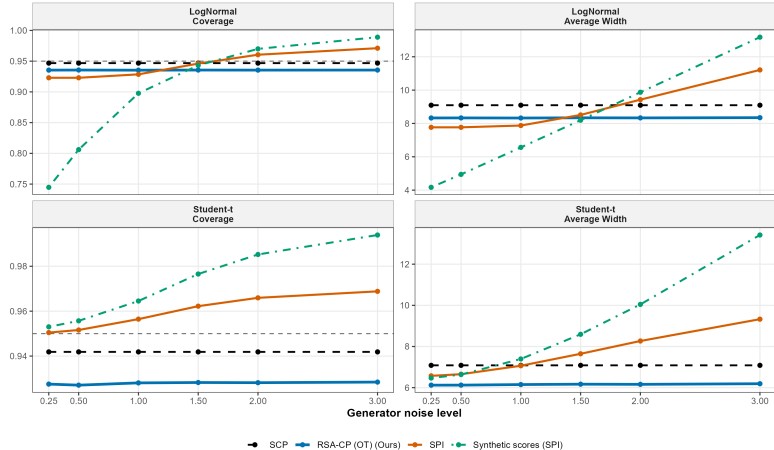

*Figure 6.* Simulation study: Coverage levels and widths of prediction sets constructed using our RSA-CP (OT) in two different noise regimes with increasing noise levels.

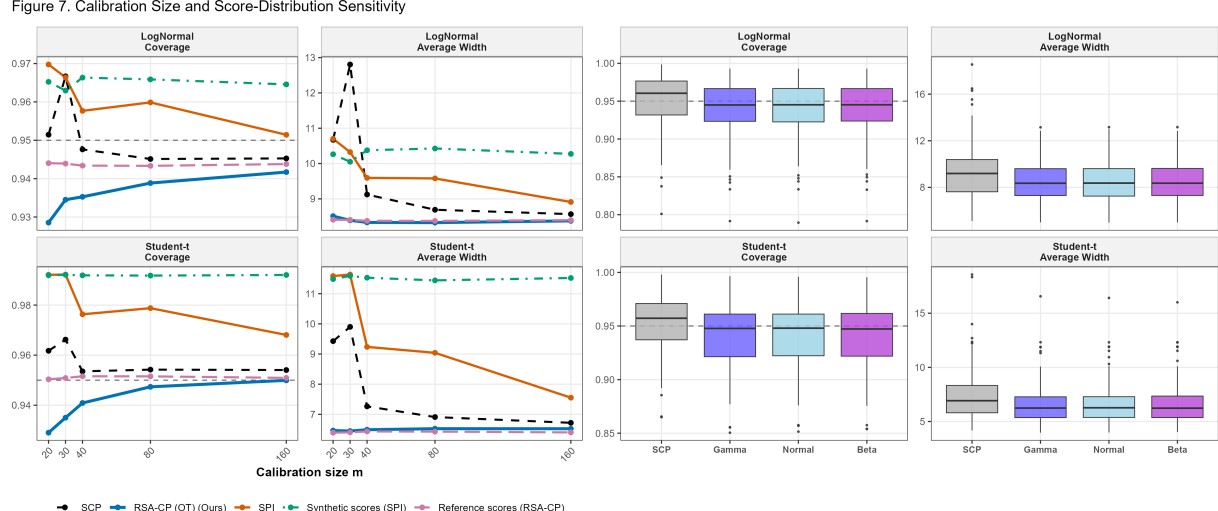

*Figure 7.* Simulation study: Left: Coverage levels and widths of different prediction sets as a function of calibration size (denoted as $N_{cal}$ in the plots). Right: Coverage levels and widths of different prediction sets in two noise regimes.

All experiments are conducted in a challenging low-calibration regime with $m = 20$ real calibration points and target miscoverage level $\alpha = 0.05$. We fix the window level at $\beta = 0.4$ to balance robustness and efficiency, and vary the number of reference scores over $N \in 20, 50, \ldots, 3000$ to study the effect of increasing rank resolution on stability. Results are averaged over 100 independent trials.

# G. Additional Real Data Experiments

## G.1. ImageNet classification

In these experiments, reference scores are generated from simple parametric reference distributions estimated from the observed calibration scores. Specifically, we model the reference score distribution using a Beta family, with parameters estimated via maximum likelihood estimation (MLE) from the $m$ observed calibration scores. This provides an approximation to the underlying score distribution while remaining computationally lightweight. Throughout the experiments, we fix $\beta = 0.4$.

For ImageNet classification experiments, candidate-label evaluation follows the standard conformal prediction protocol

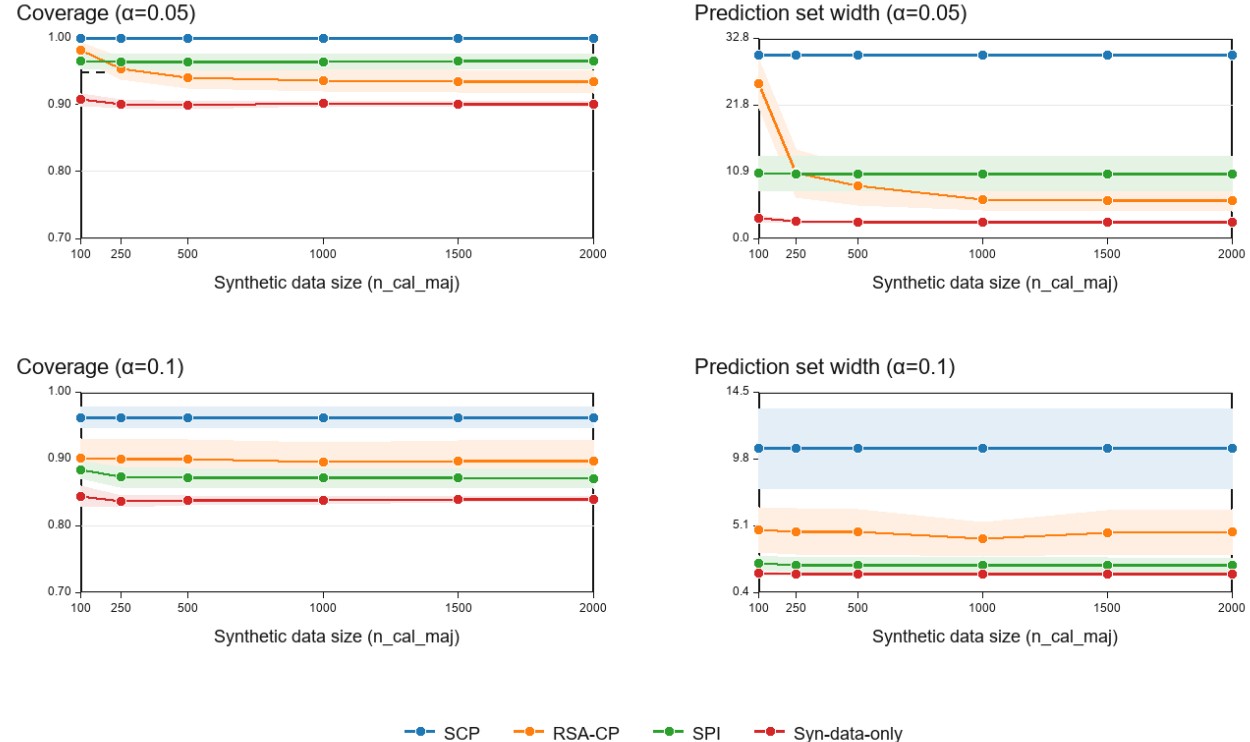

*Figure 8.* Image Classification: Coverage levels and widths of prediction sets constructed using SCP, RSA-CP(OT), SPI, and Synthetic data only methods for $\alpha = 0.05, 0.10$, calibration size 15 with increasing augmented score sizes from 100 to 2000.

used in SCP and SPI. For each candidate label $y$, we compute the corresponding nonconformity score $s(X, y)$ and apply the RSA-CP inclusion rule in the same manner as standard conformal prediction. All methods use identical pretrained models, features, logits, and preprocessing pipelines, with no additional hyperparameter tuning beyond calibration. For SPI, the synthetic score generator follows the original implementation and protocol provided in the SPI framework (Bashari et al., 2026).

### G.2. Regression on the MEPS dataset

We evaluate RSA-CP on the Medical Expenditure Panel Survey (MEPS) regression benchmark, following the experimental protocol of the SPI framework (Bashari et al., 2026).

For all methods, we use conformalized quantile regression (CQR) scores constructed from pretrained quantile regression prediction intervals provided in the SPI repository (available at `https://github.com/Meshiba/spi`). The experiments are conducted separately across four age groups: 0–20, 20–40, 40–60, and 60+. Following the small-sample setting considered in SPI, the real calibration size is fixed at $m = 15$.

We compare four approaches: (i) standard split conformal prediction (SCP) calibrated only on real minority scores, (ii) RSA-CP (OT) with score-level reference augmentation, (iii) SPI, and (iv) CP using only synthetic scores.

In case of SPI, we follow the protocol outlined in (Bashari et al., 2026). Specifically, MEPS panel 19 is used to train the underlying regression model, panel 20 serves as the reference dataset, and panel 21 is treated as the target minority dataset for calibration and testing. This setup reflects a realistic deployment scenario in which large historical populations are available as auxiliary reference data, while only a small number of samples are available from the current target population.

For RSA-CP, we generate additional reference scores from the reference distribution (MEPS panel 20). In our experiments, we vary the number of reference scores from $N \in 250, 500, \dots, 2500$ while keeping the real calibration size fixed at $m = 15$. The experiments are repeated over 50 random splits, and we report empirical coverage and average prediction interval length

evaluated exclusively on minority-group test observations. We consider target miscoverage levels $\alpha \in 0.05, 0.10$.

The results demonstrate that RSA-CP consistently achieves coverage close to the nominal level while often producing shorter prediction intervals than SCP. Similar to the observations reported for SPI, the standard conformal baseline exhibits high variability and can become overly conservative in the small-sample regime, whereas generated-only methods may perform well empirically but lack finite-sample validity guarantees. In contrast, RSA-CP effectively leverages additional reference information through score transport while retaining rigorous finite-sample coverage guarantees.

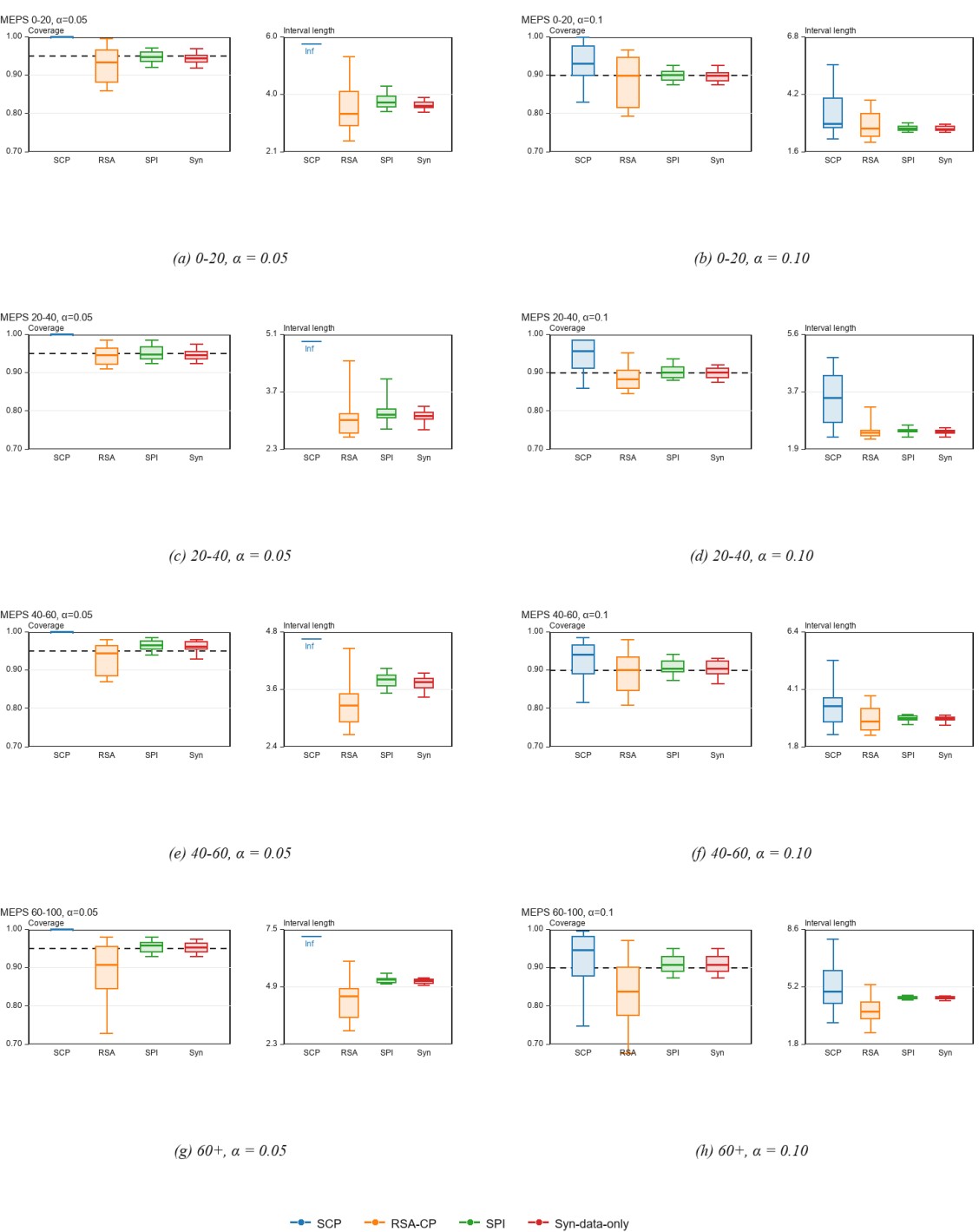

*Figure 9.* MEPS regression experiments across different age groups and target miscoverage levels. Coverage and prediction interval lengths are reported for SCP, RSA-CP (OT), SPI, and synthetic-only calibration methods. The real calibration size is fixed at $m = 15$, while the number of reference/synthetic scores is fixed at $N = 1000$.

# H. Sensitivity to alignment mismatch

We conduct an additional score-level sensitivity experiment to isolate the effect of alignment error in RSA-CP (Theorem 3.4). Real calibration scores, reference scores, and test scores are generated from the same distribution, so the correct population alignment is the identity map. We then deliberately perturb the alignment map away from identity through a monotone tail distortion $T_\delta$, where $\delta = 0$ corresponds to perfect alignment and larger $\delta$ induces increasing mismatch between the transformed real scores and the reference scores. In particular,

$$T_\delta(s) = s + \delta \, \frac{s^2}{\hat{\sigma}_s + |s| + \varepsilon}, \qquad \delta \geq 0,$$

where $\hat{\sigma}_s$ is the interquartile range of the real calibration scores and $\varepsilon > 0$ is a small numerical constant. Thus $T_0(s) = s$, while larger values of $\delta$ increasingly inflate upper-tail scores and hence increase the discrepancy between the transformed real-score distribution $T_\delta \# P$ and the reference score distribution. We set $m = 15, N = 1000, n_{test} = 1000, \beta = 0.40, \alpha = 0.05$ and vary $\delta \in \{0.0, 0.05, 0.10, 0.20, 0.30, 0.40, 0.50, 0.75, 1.00\}$

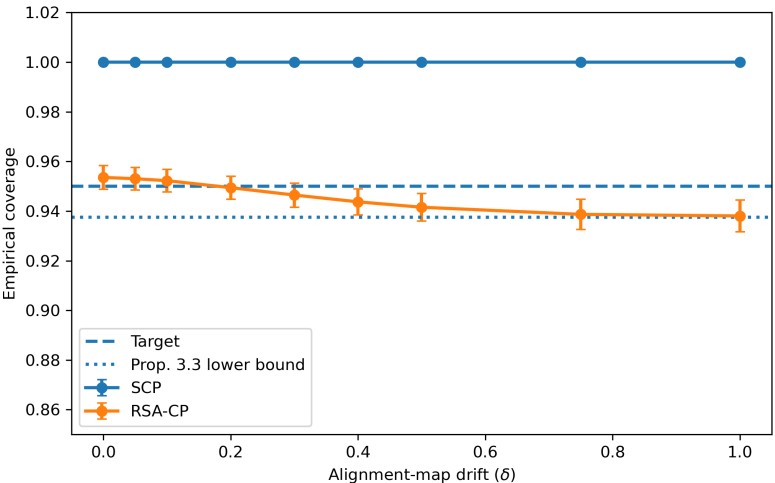

*Figure 10.* **Sensitivity to alignment mismatch under lognormal scores:** Real calibration scores, reference scores, and test scores are generated from the same lognormal score distribution, so the correct population alignment is the identity map. We then introduce controlled alignment mismatch through $T_\delta(s) = s + \delta s^2 / (\hat{\sigma}_s + |s| + \varepsilon)$.

Figure 10 reports the lognormal setting as in our section 4. The results show that RSA-CP remains valid under increasing alignment mismatch: coverage stays above the Proposition 3.3 lower bound, while the method becomes more conservative as the perturbed alignment map moves farther from identity. At $\delta = 0$, RSA-CP produces finite informative prediction sets even though SCP is vacuous for $m = 15$ and $\alpha = 0.05$. As $\delta$ increases, the augmented-rank refinement is used less frequently and RSA-CP moves toward the conservative lower-bound regime. Empirical coverage remains within the finite-sample bounds predicted by Proposition 3.3.

