# OpenReview forum: "RSA-CP: Efficient Conformal Prediction in Small-Sample Regimes via Random Score Alignment"
_ICML.cc/2026/Conference — ICML 2026 regular_

### Official Review · Reviewer_XJAm · 2026-02-18

**Soundness:** 3
**Presentation:** 4
**Significance:** 3
**Originality:** 4
**Overall Recommendation:** 5
**Confidence:** 4

**Summary:**

This paper proposed RSA-CP, an algorithm so called "random score alignment" that addresses the inefficient/non-informative conformal prediction (CP) (overly wide interval width) in small sample regimes. The idea behind is interesting and smart, where RSA-CP uses a calibration technique by aligning real scores with a high-resolution reference score distribution, avoiding estimating parameters or computation from the training data or generating new data.

I feel this paper provides new insights and offers a helpful avenue in CP problems for future investigation, especially there was limited effort in addressing the small sample size and conservative prediction issues in CP literature. The paper also has solid results in theory. I am currently hesitate to give a full acceptance (score 5 or 6), since I still have a few minor suggestions and comments.

**Compliance With Llm Reviewing Policy:**

Affirmed.

**Final Justification:**

My concerns are well addressed. The author's response is good and I think they should incorporate the suggested changes in the camera-ready paper, as they show how these points can be revised/edited clearly. These changes are mostly minor, and no major modifications are needed. Therefore, I raised my score to 5, and I didn't see a reason to reject this work. This could be a good and interesting addition to your conference.

**Key Questions For Authors:**

None. See above.

**Limitations:**

See above.

**Strengths And Weaknesses:**

**Strengths**

Interesting idea, original perspective, good theoretical investigations, and nice presentations for paper flow, algorithm, theorems, simulation and data applications.

**Weaknesses/Questions/Comments**

I have a major question for the authors: how can you decide or guide the selection of the reference distribution in practice and in general? Could you specify more details of finding the reference score in your data application? I didn't find any details even in Appendix E. And how would you generalize this (or give a general instruction) to other real applications? Or at least, the authors should discuss very explicitly what are the appropriate real data situations the framework can be applied to.

In Section 2.2, you mentioned when $m$ is small, the empirical distribution become coarse. Perhaps it is better to connect this with the inequality right before the Section 2.2 title, so that reminds why the coverage is less stable, because it is controlled by $1/(1+m)$ in general.

Simulation results show that the method achieve a great balance on coverage, efficiency and computational time. But in some scenarios, the proposed method has slight under-coverage to the nominal level 0.95, what is the main reason for this? As can be seen, in Figure 2, when $N$ becomes larger, the coverage becomes very stable around 0.93, not 0.95. Is this related to any theoretical results and should be regarded as expected in practice? Clarifying this would be important.

As the authors mentioned, they used a misspecified generator with a rare-shock component for the reference scores. Can authors possibly add a small experiment for sensitivity check for showing how coverage/performance changes as the reference scores quality varies?

The authors should also explicitly discuss how to apply their framework to small sample clinical trial data analysis, because I think there are very direct connections. Clinical trials for rare disease often have small sample sizes, especially limited control participants due to ethics concerns. If I want to provide a prediction interval for the individual treatment effect for a new patient visiting my clinic, and I only have limited trial data, how would your framework be possibly applied? One recent trend is leveraging the external controls data, which is also recommended by US FDA's guideline. Can you possibly get the reference score from the external controls and apply your method? Reference:

- Zhu, Ke, Shu Yang, and Xiaofei Wang. "Enhancing statistical validity and power in hybrid controlled trials: A randomization inference approach with conformal selective borrowing." arXiv preprint arXiv:2410.11713 (2024).
- Gao, Chenyin, et al. "Improving randomized controlled trial analysis via data-adaptive borrowing." Biometrika 112.2 (2025): asae069.

In addition, discussion in Introduction or Discussion section on literature for integrating multiple data sources in CP would be helpful, which may also be alternatives for efficiency gains but they need to consider the covariate and/or label shifts issue:

- Liu, Y., Levis, A. W., Normand, S. L., & Han, L. (2024). Multi-source conformal inference under distribution shift. Proceedings of machine learning research, 235, 31344.

---

> ### Author Rebuttal · Authors · 2026-03-31
>
> We thank the reviewer for the positive assessment and thoughtful suggestions. We address the key points below.
>
> ---
>
> **Choosing the reference score distribution**
>
> Proposition 3.3 and Theorem 3.4 should be read together as a practical guide. The key insight is that the choice of reference distribution affects *efficiency* but not *validity*: the distribution-free bounds in Proposition 3.3 hold for any reference distribution, so no choice of reference distribution can invalidate the coverage guarantee. Theorem 3.4 then quantifies how much the coverage can be tightened around $1-\alpha$: this tightening improves as the TV discrepancy between the transported real-score distribution and the reference distribution decreases.
>
> In practice, we recommend the following approach. Choose a simple parametric family (e.g., Beta distributions for scores bounded in $[0,1]$ as done in our real data experiment, or log-normal distributions for positive-valued scores) and estimate the parameters from the observed calibration scores via maximum likelihood. This provides a reasonable approximation to the true score distribution while maintaining stability. The OT alignment step then corrects for any residual mismatch between this parametric approximation and the true score distribution. When domain knowledge or auxiliary data are available, for example, historical data from related tasks, or external control cohorts in clinical applications, reference scores can be constructed from those sources to improve alignment further. The validity guarantee of Proposition 3.3 holds regardless, providing a safety net even when the reference distribution is poorly chosen.
>
> **Coverage below the nominal level in experiments**
>
> The stabilization of coverage around 0.93 rather than 0.95 in Figure 2 (for large $N$) reflects the fundamental certifiable coverage-efficiency tradeoff in small-sample conformal prediction, and is not an artifact or failure of RSA-CP. There are two contributing factors.
>
> *Factor 1: Fixed $\beta$ and the certified lower bound* In our simulations, we use a fixed $\beta = 0.4$ without enforcing the certified condition $L(\beta, m) \geq 1-\alpha$. For $m = 20$ and $\alpha = 0.05$, the distribution-free lower bound $L(\beta, m)$ at this $\beta$ may fall below $0.95$. The observed coverage of approximately 0.93 is consistent with this: it is above the actual lower bound achievable at the chosen $\beta$, modulated by the alignment quality captured by Theorem 3.4.
>
> *Factor 2: Residual distributional mismatch* Even as $N \to \infty$, if the OT alignment is imperfect due to the deliberately misspecified reference generator (with rare-shock component), the TV discrepancy term in Theorem 3.4 does not vanish entirely for finite $m$. This creates a persistent gap between the observed coverage and the nominal level $1-\alpha = 0.95$.
>
> Crucially, as $N \to \infty$ with $m$ fixed, the rank grid refinement saturates and the dominant source of the coverage gap
> shifts entirely to the residual alignment error and the fixed-$\beta$ lower bound. This explains why coverage stabilizes
> rather than continuing to improve.
>
> For other asymptotic behaviors, please see our response **Asymptotic regime** to the **Reviewer qmc2**.
>
> **Sensitivity to reference score quality** We will add a sensitivity experiment in the revision varying the shock probability in $\{0, 0.01, 0.02, 0.03, 0.04\}$, where $0$ corresponds to a perfectly specified reference generator and $0.05$ corresponds to the baseline misspecification used in the main simulation study. We expect that RSA-CP (OT) empirical coverage gets closer to the target level as shock probability decreases while efficiency gains persist, with the distribution-free bound of Proposition 3.3 providing a floor. This experiment will directly illustrate the practical role of Theorem 3.4.
>
> **Application to small-sample clinical trials**
>
> We thank the reviewer for this suggestion. Small-sample clinical trials are a natural application of RSA-CP, where standard SCP is ineffective due to limited calibration data. RSA-CP addresses this by leveraging reference scores from external controls, historical trials, or registry data, which are increasingly used and encouraged by regulatory agencies (e.g., FDA). Unlike existing borrowing methods that operate at the data or model level (recent work on selective and adaptive borrowing (Zhu et al., 2024; Gao et al., 2025)), RSA-CP borrows *rank resolution*, providing finite-sample validity guarantees. We will add a discussion of this application in the revision.
>
> We note that extending RSA-CP to settings with distribution shift is nontrivial, as current guarantees rely on exchangeability (in particular, Beta Binomial law for conditional ranks). This is an important direction for future work.
>
> **Presentation**  We will revise Section 2.2 to explicitly connect it with the preceding inequality. Thank you for the suggestion.
>
>
> We hope this addresses the reviewer’s suggestions.

---

> > ### Author Rebuttal · Reviewer_XJAm · 2026-03-31
> >
> > The authors' response is satisfactory. I believe they will include proposed changes.

---

> > > ### Author Response · Authors · 2026-04-02
> > >
> > > We are glad that our response has addressed all of your concerns. We thank the reviewer for the thoughtful questions and suggestions, which have helped clarify and strengthen the contributions of our work. We will incorporate all the proposed changes which will improve the presentation of the paper.

---

### Official Review · Reviewer_xUj5 · 2026-03-11

**Soundness:** 2
**Presentation:** 3
**Significance:** 3
**Originality:** 2
**Overall Recommendation:** 4
**Confidence:** 2

**Summary:**

The authors introduce a split conformal prediction (SCP) method for data-scarce scenarios, called RSA-CP, which uses Random Score Alignment (RSA). This approach allows them to produce tighter CP intervals than SCP while still achieving finite-sample coverage guarantees. Indeed, SCP intervals with a small calibration set often yield overly conservative prediction sets due to low quantile resolution. In the existing literature, one way to address this issue is to generate additional nonconformity scores to augment the calibration set, but this requires assumptions on the true score distribution to maintain finite-sample marginal validity. In contrast, the authors propose augmenting score ranks rather than numerical score values to achieve the desired coverage level in the resulting prediction sets, using a Beta-binomial distribution over the ranks. They redefine prediction sets based on conditional ranks and calculate lower and upper bounds. To further ensure coverage guarantees, they propose an Optimal Transport (OT)-based approach to map true scores to a reference score distribution specified by the user. Finally, they implement RSA-CP on both simulated and real datasets and demonstrate improvements in execution time and prediction interval width compared to SCP and an existing method for small calibration samples, called SPI.

**Compliance With Llm Reviewing Policy:**

Affirmed.

**Final Justification:**

The authors have adequately addressed my concerns regarding some theoretical and experimental aspects of the paper, and I have therefore upgraded my score accordingly. However, as mentioned in my initial review, I do not consider myself an expert in Uncertainty Quantification (and Conformal Prediction), so I will keep my confidence score unchanged, as I could not verify all the mathematical proofs.

**Key Questions For Authors:**

1) In practice, how should one choose the reference score distribution to attain the desired coverage level, given the results of Theorem 3.4? Is it really achievable in practice to have a distance between transported score and reference score distributions close to 0?
2) From the figures in the experimental study, it appears that the desired coverage level is never fully attained using RSA-CP. Could the authors discuss this limitation more explicitly in the paper?
3) Is there a specific reason why the authors chose to compare RSA-CP to only one existing method addressing small calibration sets (SPI), while other methods appear to exist in the literature?
4) Could the authors explain the difference in the observed results between Figure 2 (simulation study) and Figure 3 (real data)? The results seem to be opposite: for real data, RSA-CP achieves better coverage but longer intervals compared to SPI, whereas in the simulation study, the trend appears different.
5) I did not fully understand the remark regarding $n_{cal}$. Could the authors provide more detail on why RSA-CP converges to the desired coverage level as $n_{cal}$ increases, and how this compares to SPI?

**Limitations:**

yes

**Strengths And Weaknesses:**

1) Soundness

It is worth noting that, as I have just started working on Uncertainty Quantification using Conformal Prediction (CP), I did not have time to redo the proofs of the main results in the paper. However, it appears that I have understood the details of the introduced method, RSA-CP. In addition,
- At the beginning of the paper, the authors assume nonconformity scores are continuous to avoid ties in the observed scores. Is this a commonly used assumption in CP methods?
- In Proposition 3.2, there appears to be an error: I believe the authors refer to Equation (6) rather than Equation (8).
- If I understand Section 3.5 and Theorem 3.4 correctly, there is no guaranteed existence of a $\beta$ that will achieve marginal validity at the desired miscoverage level $\alpha$. When using the OT mapping, this is ensured only when the distance between the transported scores and the reference score distribution is exactly matched, which seems to be an important limitation.
- Page 6, line 323 in the left column: it should be $\Tilde{Q}$ rather than $\Tilde{P}$.

2) Presentation

Overall, I find the article well-structured. The motivation and contribution are clearly stated in the Introduction, which makes the rest of the article easier to understand. However, in Section 2.2, a simple illustrative example of the addressed concern (small calibration set) could be added to improve clarity, that demonstrates potential failure of traditional SCP.

3) Significance

The practical utility of the approach is clearly demonstrated: in many applications, labeled examples can be sparse, resulting in a small calibration sample for CP procedures. The idea of developing an SCP approach based on score ranks rather than numerical scores clearly reduces execution time while producing tighter CP intervals compared to existing methods that rely on simulating additional calibration nonconformity scores.

4) Originality

I find the idea of using score ranks rather than numerical values very interesting. The authors provide promising theoretical results for RSA-CP, with improvements to address some limitations of their initial approach by using Optimal Transport (OT) theory in the univariate case, making the method easy to implement compared to multivariate OT. Moreover, the authors situate their work well within the existing CP literature for small calibration sets, both in the Introduction (and Appendix A). However, in Section 4 (experimental study), they only compare RSA-CP to one existing approach, while multiple other methods seem to exist.

---

> ### Author Rebuttal · Authors · 2026-03-31
>
> We thank the reviewer for the careful reading. We clarify the key points below.
>
> ---
>
> **Reference distribution and OT alignment**
>
> Proposition 3.3 and Theorem 3.4 together clarify that the reference distribution affects *efficiency* but not *validity*. The distribution-free bounds in Proposition 3.3 hold for any reference distribution, while Theorem 3.4 shows that coverage tightens toward $1-\alpha$ as the TV discrepancy between aligned real scores and the reference distribution decreases. In practice, we use simple parametric families (e.g., Beta distribution for $[0,1]$ bounded scores) fitted via MLE from calibration scores, with OT alignment correcting residual mismatch. When available, domain knowledge or auxiliary data can further improve alignment. In finite samples, the TV distance cannot be zero, explaining slight under-coverage. However, empirical coverage remains within the finite-sample bounds of Proposition 3.3, ensuring validity despite imperfect alignment.
>
> **Coverage below the nominal level in experiments**
>
> This behavior has a clear theoretical explanation. In our experiments, we use a fixed $\beta = 0.4$ rather than the certified $\beta$ from Algorithm 2. As discussed above, this means the distribution-free lower bound $L(\beta,m)$ may fall below $1 - \alpha$ for chosen $\beta$. The OT alignment step then works to close this gap by reducing the TV discrepancy in Theorem 3.4, but when alignment is imperfect, a small deviation from the nominal level remains. Importantly, this is a deliberate experimental choice, not a failure of the method. We fixed $\beta$ precisely to study the behavior of RSA-CP when certification is not enforced, and to illustrate the role of OT alignment in improving coverage in this regime. As shown in Figure 4, coverage converges toward $1-\alpha$ as $m$ increases, which is exactly the behavior predicted by Theorem 3.4.
>
> **Comparison with existing methods: why only SPI?**
>
> We chose SPI as the primary baseline because it is the most directly comparable recent method: it addresses small calibration sets, shares the same oracle motivation (augmentation for rank resolution), and provides finite-sample coverage guarantees. Other approaches either lack finite-sample guarantees (relying on asymptotics) or assume auxiliary data are drawn from the same distribution as the calibration data, which RSA-CP does not require. As such, their guarantees are not directly comparable. We discuss these methods in Appendix A and will clarify this positioning in the main text. We will also add a regression benchmark (MEPS dataset following SPI paper) to demonstrate generality across tasks.
>
>
> **Difference between Figure 2 (simulation) and Figure 3 (real data)**
>
> The trends in Figures 2 and 3 reflect different alignment regimes, as captured by Theorem 3.4 and Proposition 3.3.
>
> In Figure 2 (simulation), we deliberately use a misspecified reference distribution with rare shocks to stress-test robustness. Here, OT alignment is crucial: RSA-CP (OT) corrects mismatch and achieves near-target coverage, while SPI is unstable, exhibiting overcoverage in one regime and undercoverage in another.
>
> In Figure 3 (real data), alignment is more favorable due to the fitted Beta reference distribution. RSA-CP achieves near-nominal coverage with smaller sets than SCP, while SPI attains smaller sets but suffers from undercoverage in this extremely small calibration regime ($n_{\text{cal}}=15$).
>
> Thus, the figures are complementary, not contradictory: they illustrate RSA-CP under poor vs. favorable alignment.
>
> **Behavior as calibration size $n_{cal}$ increases**
>
> For fixed $N$, RSA-CP approaches the target coverage $1-\alpha$ as $n_{cal}$ increases (Figure 4), consistent with Theorem 3.4. As $n_{cal}$ grows: (i) the augmented rank grid $1/(n_{cal}+N+1)$ refines, reducing conservatism; and (ii) the OT map converges to its population limit $T^{\*}$ with $T^{\*} P=Q$, driving the TV discrepancy term to zero. Both effects push coverage toward $1-\alpha$.
> In contrast, SPI uses a fixed threshold $\lceil(1-\alpha)(N+1)\rceil$ that does not adapt to $n_{cal}$, so coverage stabilizes below target as real data dominate. We will clarify this mechanism and link it explicitly to Theorem 3.4 in the revision.
>
> **Notation and presentation issues**  Thank you for pointing out the typo on Page 6, line 323. We will carefully audit the manuscript and correct all such issues in the revision.
>
> ---
>
> We hope this clarifies the reviewer’s concerns and are happy to provide further details if needed.

---

> > ### Author Rebuttal · Reviewer_xUj5 · 2026-04-02
> >
> > The additional clarification has addressed my earlier concerns, so I will upgrade my score.

---

### Official Review · Reviewer_LNbM · 2026-03-13

**Soundness:** 2
**Presentation:** 2
**Significance:** 3
**Originality:** 3
**Overall Recommendation:** 4
**Confidence:** 3

**Summary:**

The paper addresses conformal prediction with very small calibration sets, where split conformal prediction becomes coarse and inefficient due to limited rank resolution. It proposes RSA-CP, which augments the calibration process with many auxiliary reference scores and uses a rank-based construction to refine prediction-set decisions. A central component of the method is a Beta-Binomial characterization of the augmented rank, which yields finite-sample coverage bounds and is transferred from rank space to score space through monotone score alignment, implemented mainly via OT barycentric alignment. Empirically, the authors report improved efficiency and stability over standard split conformal prediction, and lower computational cost than synthetic-data-based alternatives, in simulations and a small-calibration image classification benchmark.

**Compliance With Llm Reviewing Policy:**

Affirmed.

**Final Justification:**

The rebuttal adequately addressed my main concerns, especially by clarifying the implementation details, resolving the $j^*$ inconsistency, and better explaining the relationship to SPI.

**Key Questions For Authors:**

- Which definition of the cutoff index $j^\*$ is correct in the implemented method: $\lceil(1-\alpha)(N+m+1)\rceil$ from Section 3.3 or $\lceil(1-\alpha)(N+1)\rceil$ from Algorithm 1?
- Can you report the actual theoretical lower/upper coverage bounds $L(\beta,m)$ and $U(\beta,m)$ alongside empirical coverage in the experiments?

**Limitations:**

The paper does not discuss limitations in enough detail. In particular, it should more clearly explain the practical consequences of relying on reference-score quality, possible failure under distribution mismatch and the fact that the main theory gives coverage bounds that don't seem to be known a priori so as to ensure sufficient coverage.

**Strengths And Weaknesses:**

The paper identifies a real weakness of split conformal prediction in the small-$m$ regime and proposes a method that is well matched to that problem.
The motivation, rank-based derivation, score-space mapping, and alignment step follow a coherent sequence.

However, I have a few concerns:
- The experimental section seems to rely on a fixed $\beta$, so it is not fully clear whether the practical results correspond to the certified version of the method.
- There is at least one potentially consequential inconsistency in the algorithmic specification. In Section 3.3 the nominal augmented cutoff is defined as $j^\*=\lceil(1-\alpha)(N+m+1)\rceil$, but Algorithm 1's Step 5 sets $j^\*\leftarrow \lceil(1-\alpha)(N+1)\rceil$.
- The empirical validation is relatively narrow. The real-data evidence is a single ImageNet-derived benchmark borrowed from SPI, and the simulation study uses one calibration size in the main text ($m$=20) with further sensitivity in the appendix. This is not enough to establish broad robustness across tasks, score functions, or conformal settings.
- The relationship to SPI is somewhat unclear at the conceptual level. The appendix acknowledges that SPI also uses a score transporter / empirical quantile mapping, but the paper does not sufficiently sharpen where RSA-CP is fundamentally different versus where it is a stripped-down special case that removes feature-conditioned synthesis and retools the guarantee around reference-score ranks.
- Some implementation details remain underspecified, especially for the real-data experiments: how reference scores are sampled there, whether the same $\beta$ is used throughout, how candidate-label evaluation is performed for large label spaces, and whether all baselines were tuned comparably.

Additionally, the manuscript has a nontrivial number of writing and notation issues such as inconsistent theorem/proposition references ("Theorem 3.3" vs "Proposition 3.3") and and the $j^\*$ inconsistency noted above.

Overall, the paper has a technically interesting core idea, and the Beta-Binomial rank analysis appears meaningful. However, there are still important concerns about whether the experimental results correspond to the certified version of the method, whether theory and implementation fully match, and whether the empirical evaluation is broad enough to support strong claims.

---

> ### Author Rebuttal · Authors · 2026-03-31
>
> We thank the reviewer for thorough feedback. We address the raised concerns below.
>
> ---
>
> **Certified vs empirical version (Choice of fixed $\beta$ in experiments)**
>
> In the experiments, we use a fixed value $\beta = 0.4$ rather than selecting $\beta$ via Algorithm 2 (the certified selection procedure). When such $\beta$ exists, the certified version provides explicit finite-sample guarantee. The practical version with fixed $\beta$ is still governed by the bounds in Proposition 3.3 and Theorem 3.4. We chose fixed $\beta$ for two principled reasons. First, it enables a fair and controlled comparison with SPI, which also uses fixed hyperparameter settings. Second, it creates a setting where the certified condition $L(\beta,m) \geq 1 - \alpha$ may fail, making the OT alignment step essential - this directly illustrates the practical behavior described in Section 3.5.
>
> In the revision, we will report the bounds $L(\beta, m)$ and $U(\beta, m)$ alongside empirical coverage in all figures, so the reader can directly verify the relationship between the certified bounds and observed performance (In particular, for $m = 20, \beta = 0.4, N = 1000$, we have $L(\beta,m) = 0.9048$ and $U(\beta,m) = 1.0$).
>
>
> **How does RSA-CP differ from SPI?**
>
> Both methods share the same oracle motivation (augmenting scores to improve rank resolution), but differ fundamentally in mechanism and guarantees.
>
> *Rank-based vs synthetic-data-based augmentation:*
> RSA-CP develops a purely *rank-based framework*, modeling augmentation directly via the Beta-Binomial conditional rank law. It does not require synthetic datapoints or learned generators. In contrast, SPI relies on feature-conditioned synthetic data and empirical quantile mapping. RSA-CP instead uses simple reference scores (e.g., Beta draws), making it computationally lightweight and generator-free.
>
>  *Threshold construction:*
> RSA-CP uses $j^\star=\lceil(1-\alpha)(m+N+1)\rceil$, defined on the *joint augmented rank scale* of real and reference scores. In contrast, SPI uses $\lceil(1-\alpha)(N+1)\rceil$, calibrated only on synthetic scores. Our threshold arises from conditional rank localization and explicitly accounts for real calibration data, whereas SPI remains anchored to the synthetic distribution.
>
> *Consequences:*
> This distinction leads to different asymptotic behavior. As $m$ increases (with fixed $N$), RSA-CP coverage converges to $1-\alpha$ (Theorem 3.4), since the joint rank grid refines and the discrepancy term vanishes. In contrast, SPI coverage stabilizes below target, as its threshold does not adapt to $m$. This difference is evident in Figure 4 and reflects a fundamental gap between rank-based augmentation (RSA-CP) and synthetic-score calibration (SPI).
>
>
> **Empirical scope**
>
> We used the ImageNet-derived benchmark from SPI to enable a controlled comparison under identical data, model, and protocol, ensuring differences are attributable to the calibration mechanism. In simulations, $m=20$ highlight the small-sample regime, with sensitivity over $m,N,\alpha$ results in the appendix. Real data experiment already shown varying $m$ choices. Since RSA-CP is rank-based and task-agnostic, improvements arise from calibration rather than task-specific modeling. In the revision, we will expand experiments by adding a regression benchmark (MEPS, following model choices in SPI paper) to demonstrate cross-task performance.
>
>
> **Implementation details**
>
> We will expand this section in the revision. In real-data experiments, reference scores are sampled from a Beta distribution with MLE parameters estimated from the $m$ calibration scores. We use $N=1000$ and $\beta=0.4$ throughout. For ImageNet, candidate-label evaluation follows standard conformal practice: for each label $y$, we compute $s(X_{m+1},y)$ and apply the RSA-CP rule, identical to SCP and SPI.
> All methods use identical models, features, and preprocessing, with no additional tuning beyond calibration; SPI’s generator follows the original protocol.
>
>
> **Notation and presentation issues** Thank you for pointing this out. The correct cutoff is $j^\star=\lceil(1-\alpha)(m+M+1)\rceil$, as defined in Section 3.3. The expression in Algorithm 1, $j^\star=\lceil(1-\alpha)(M+1)\rceil$, is a typographical error. All experiments use the correct definition. We will correct this and other notational inconsistencies in the revision. These issues do not affect any theoretical results or empirical findings.
>
> **Limitations and coverage guarantees** RSA-CP relies on alignment only for *efficiency*, not validity. Under severe mismatch, the coverage bounds in Theorem 3.1 still holds. The  bounds in Theorem~3.1 are fully computable for any $(m,N,\beta)$ and are distribution-free. In practice, $(L(\beta,m),U(\beta,m))$ can be evaluated before deployment, allowing $\beta$ to be chosen to ensure desired coverage.
>
> ---
>
> We hope this addresses the reviewer’s concerns and are happy to clarify further if needed.

---

> > ### Author Rebuttal · Reviewer_LNbM · 2026-04-02
> >
> > I'd like to thank the authors for their response. The rebuttal adequately addressed my main concerns, especially by clarifying the implementation details, resolving the $j^*$ inconsistency, and better explaining the relationship to SPI. I am therefore increasing my score accordingly.

---

### Official Review · Reviewer_qmc2 · 2026-03-13

**Soundness:** 3
**Presentation:** 3
**Significance:** 3
**Originality:** 4
**Overall Recommendation:** 5
**Confidence:** 4

**Summary:**

The paper proposes a conformal prediction method that incorporates synthetic samples into the conformity score in an effort to improve finite-sample efficiency while retaining the usual finite-sample coverage guarantee. The motivation is an oracle argument: if one could augment the calibration set with additional draws from the true conformity-score distribution, then the effective calibration sample size would increase and the resulting intervals could become tighter. The proposed method aims to approximate this idea by generating $M$ synthetic samples and using them to construct a modified conformity score. The paper develops a theoretical framework for this construction and shows that the resulting procedure still enjoys finite-sample marginal coverage. It also identifies a practical issue - namely, that the nominal confidence level may not exactly achieve the target coverage level - and proposes a score-alignment procedure based on optimal transport to address this. The empirical results, across a range of datasets and models, suggest that incorporating synthetic samples can lead to shorter intervals while maintaining or improving coverage relative to standard conformal prediction.

**Compliance With Llm Reviewing Policy:**

Affirmed.

**Final Justification:**

The authors addressed my concerns, and I appreciate the additional intuition and discussion they propose to add in the revision. Accordingly, I have raised my score.

**Key Questions For Authors:**

# Questions
My questions are follow-ups to the weaknesses discussed above.


- Why should the proposed procedure be more efficient than standard conformal prediction, in the sense of yielding shorter intervals?

- The oracle argument assumes access to exchangeable or i.i.d. draws from the true conformity-score distribution. Why should the conformity score constructed from $M$ synthetic samples inherit the efficiency benefits of that oracle procedure?

- What should one expect when $M$ is small versus large?

- If the $M$ synthetic samples approximate the oracle setting as closely as possible, how should the resulting efficiency compare with that of the oracle procedure?

- Can the method retain the oracle efficiency gain, or is there necessarily a tradeoff?

- If $n \to \infty$ with $M$ fixed, do the synthetic samples have any first-order effect on efficiency?

- What happens when $M$ grows with $n$, and can any resulting efficiency improvement be formally characterized?

**Limitations:**

Yes

**Strengths And Weaknesses:**

# Strengths

- The paper proposes a novel conformity score that incorporates synthetic samples while still maintaining finite-sample coverage.

- The resulting procedure is relatively simple and appears computationally tractable.

- The theoretical development is mathematically well motivated and clearly connected to the proposed construction.

- The paper also identifies a limitation of the proposed approach, namely that it may not be possible to tune the confidence level to attain the desired coverage exactly. To address this issue, the authors introduce a score-alignment procedure based on optimal transport.

- The experiments are comprehensive and provide empirical evidence that the procedure can improve finite-sample efficiency while maintaining or improving coverage.

# Weaknesses



While the proposed methodology is technically impressive and the experiments suggest that incorporating synthetic samples can be practically beneficial, *the paper does not yet provide sufficient intuition or theory for when and why these efficiency gains should occur*. In particular, it remains unclear why the proposed procedure should be more efficient than standard conformal prediction, in the sense of yielding shorter intervals.

The motivation is based on an oracle argument: if one could enlarge the calibration set by \(M\) additional data points, then coverage tightness would improve at rate \(O(1/M)\). However, this oracle argument assumes access to exchangeable or i.i.d. draws from the true conformity-score distribution. By contrast, the proposed method constructs a conformity score using \(M\) synthetic samples, and the paper does not explain why this construction should inherit the efficiency benefits of the oracle procedure.

The role of \(M\) is also unclear. What should one expect when \(M\) is small versus large? If the \(M\) synthetic samples approximate the oracle setting as closely as possible, how should the resulting efficiency compare with that of the oracle procedure? Can the method retain the oracle gain, or is there necessarily a tradeoff? Some discussion of this point seems important, since such improvements would not typically come for free.

The asymptotic implications are similarly underdeveloped. If \(n \to \infty\) with \(M\) fixed, I would expect the synthetic samples to have no first-order effect on efficiency, but the paper does not address this. More generally, it would be helpful to understand what happens when \(M\) grows with \(n\), and whether any efficiency improvement can be formally characterized in that regime.

---

> ### Author Rebuttal · Authors · 2026-03-31
>
> We thank the reviewer for the thoughtful and constructive feedback. We address each concern in detail below.
>
> ---
>
> **Intuition for efficiency gains** The key limitation of SCP in small samples is coarse rank resolution: with $m$ calibration points, ranks lie on a grid of size $1/(m+1)$, forcing a coarse estimate of the $(1-\alpha)$-quantile and resulting in inflated thresholds and overly wide prediction intervals.
>
> RSA-CP addresses this by augmenting the rank structure with $M$ reference scores, refining the effective rank grid to resolution $\frac{1}{m + M + 1}$. Critically, this finer grid allows the conformal threshold to be estimated more precisely, reducing the conservatism that arises from coarse quantile increments. In this mechanism (our inclusion rule), reference scores are incorporated only when they fall within a high-probability conditional rank window derived from the Beta-Binomial law. When reference scores are well aligned with real scores, they fall within these windows frequently and actively refine the rank, reducing conservatism and narrowing intervals. When reference scores are poorly aligned (e.g., different support or scale), they rarely fall within these windows, and RSA-CP effectively reverts to SCP-like behavior. This selectivity is what allows RSA-CP to improve efficiency without sacrificing coverage validity. Importantly, once the high-probability windows are established, we no longer require real and synthetic score distributions to be the same.
>
> **When and why should RSA-CP inherit the oracle efficiency benefits?**
>
> The oracle argument assumes access to $M$ additional i.i.d. draws from the true score distribution, which refines the rank grid to $\frac{1}{m + M + 1}$ and yields efficiency gains. RSA-CP uses reference scores from a potentially different distribution; our theory clarifies when similar gains occur.
>
> - Proposition 3.3 (distribution-free finite-sample bounds that hold regardless of the reference distribution)
>
> - Theorem 3.4 (distribution-robust bounds controlled by the TV discrepancy between the transformed real score distribution and the reference distribution)
>
> Together, these yield two regimes.
>
> Case I (good alignment):The TV discrepancy term is small. In this regime, RSA-CP behaves like the oracle procedure, achieving rank resolution $\frac{1}{m + M + 1}$ and near-maximal efficiency gains.
>
> Case II (severe mismatch e.g., different supports): The TV discrepancy term becomes large and the distribution-robust bound becomes loose; reference scores rarely fall within the high-probability rank windows, and RSA-CP reverts to SCP-like behavior. Importantly, validity is always preserved via Proposition 3.3, but efficiency gains may vanish.
>
> The OT alignment step is therefore essential: it reduces the discrepancy in Theorem 3.4, moving RSA-CP toward the oracle regime, as supported empirically (Figure 2).
>
> **Role of $M$ and trade-off**
> Increasing $M$ has two effects: (i) it refines the rank grid from $1/(m+1)$ to $1/(m+M+1)$, improving quantile resolution, and (ii) it stabilizes empirical reference quantiles, improving alignment. When well aligned, both yield tighter and more stable prediction sets.
> Under misalignment, increasing $M$ does not induce spurious gains: incompatible reference scores fall outside the Beta-Binomial rank windows and are effectively ignored. Thus, efficiency gains may saturate, while validity is preserved (Proposition 3.3), making larger $M$ benign.
>
> Overall, RSA-CP retains the rank-resolution benefit without requiring i.i.d. reference samples, interpolating toward the oracle regime as alignment improves (Theorem 3.4). Empirically, performance stabilizes (Figure 7), reflecting diminishing returns once the rank grid is sufficiently fine.
>
> *Trade-off:* RSA-CP trades exact calibration at level $1-\alpha$ for improved efficiency, while retaining distribution-free bounds and recovering near-target coverage under good alignment; OT alignment mitigates this trade-off.
>
>
> **Asymptotic regime:**
>
> *Regime 1*: As the real calibration size grows $m \to \infty$ with $M$ fixed, SCP already achieves quantile resolution $\mathcal{O}(1/m)$, and the $N$ reference scores contribute a negligible fraction of the augmented rank grid. In this regime, RSA-CP does not yield first-order improvements over SCP.
>
> *Regime 2*: When $m$ and $M$ grow proportionally with $M/(m+M)\to r \in (0,1)$, the OT alignment map converges to its population limit $T^{\*}$ satisfying $T^{\*}\ P = Q$. In this regime, the TV discrepancy term in Theorem 3.4 vanishes, and RSA-CP(OT) attains exact asymptotic coverage $(1-\alpha)$, recovering the oracle. Since the inclusion rule is rank-preserving and alignment is asymptotically exact, the resulting prediction sets are asymptotically equivalent to those of the oracle augmented procedure.
>
> ---
>
> In the revised manuscript, we will add the discussion in the Appendix. We hope this addresses the reviewer's concerns, and we welcome any further questions.

---

> > ### Author Rebuttal · Reviewer_qmc2 · 2026-04-04
> >
> > The authors have addressed my concerns and I will raise my score to acceptance.

---

> > > ### Author Response · Authors · 2026-04-04
> > >
> > > We thank the reviewer for the positive reassessment and for the thoughtful feedback. We are glad our responses addressed the concerns and clarified the contribution.

---

### Decision · Program_Chairs · 2026-04-30

**Decision:**

Accept (regular)

**Comment:**

The paper proposes a conformal prediction framework that improves sample efficiency in small calibration regimes by augmenting rank information with reference scores and aligning them via optimal transport .

The submission received two accepts and two weak accepts after rebuttal. Overall, reviewers agree the paper is technically sound, and the rebuttal addressed most concrete concerns, including theoretical clarification and implementation details.

The main remaining concern is the limited empirical scope and some practical aspects (e.g., choice of reference distribution), though these were partially addressed in rebuttal.

In summary, given the overall positive assessments and resolved concerns, I recommend acceptance.